# The efficacy of dance for improving motor impairments, non-motor symptoms, and quality of life in Parkinson's disease: A systematic review and meta-analysis

Anna M. Carapellotti ⬤ *, Rebecca Stevenson, Michail Doumas

School of Psychology, Queen's University Belfast, Belfast, United Kingdom

* acarapellotti01@qub.ac.uk

## Abstract

Dance may help individuals living with Parkinson's disease (PD) improve motor and non-motor symptoms that impact quality of life (QOL). The primary aim of this systematic review of randomized controlled trials (RCTs) was to evaluate the efficacy of dance in improving motor and non-motor symptoms of PD and QOL. The secondary aims of this review were to evaluate the methodological quality of included studies by assessing risk of bias across nine categories and to inform the direction of future research. Peer-reviewed RCTs that included people living with PD at all disease stages and ages and measured the effects of a dance intervention longer than one day were included. Sixteen RCTs involving 636 participants with mild to moderate PD were eligible for inclusion in the qualitative synthesis and nine in the meta-analysis. Overall, the reviewed evidence demonstrated that dance can improve motor impairments, specifically balance and motor symptom severity in individuals with mild to moderate PD, and that more research is needed to determine its effects on non-motor symptoms and QOL. RCTs that use a mixed-methods approach and include larger sample sizes will be beneficial in fully characterizing effects and in determining which program elements are most important in bringing about positive, clinically meaningful changes in people with PD.

## Introduction

Parkinson's disease (PD) is a progressive neurodegenerative disorder characterized by both motor and non-motor symptoms that impact quality of life (QOL) [1]. The cardinal motor symptoms of PD include bradykinesia, tremor, rigidity, and postural instability [2]. Non-motor symptoms, such as mental health issues (e.g., depression), cognitive impairment, pain, and fatigue, are also prevalent among people living with PD, and these issues may have a greater impact on QOL than motor impairments [3]. As PD progresses, activities of daily living (ADL) become increasingly challenging [4], and this can lead to physical inactivity, social isolation, and increased dependence on family members and carers [5]. While current pharmacological and neurosurgical treatments can help to alleviate symptoms, these methods do not

**Funding:** This work was supported by the Thouron Award to AMC. https://www.thouronaward.org/ The funder had no role in study design, data collection and analysis, decision to publish, or preparation of the manuscript.

**Competing interests:** The authors have declared that no competing interests exist.

fully address complications such as balance problems [6] and non-motor symptoms [7], leading to a need for high-quality complimentary treatment strategies that improve QOL.

Creative activities (e.g., art therapy, singing, etc.) are increasingly being recognized as viable complementary therapies for people living with PD, and there have been powerful examples of patients using artistic expression as a means of self-management [8]. Exercise is now also known to be an important adjunct to pharmacological treatments in the management of disease progression [9] and improving aspects of mobility [10]. Exercise has also been recognized as having the potential to address non-motor symptoms, such as mood, cognitive function, and sleep disorders [11, 12].

Dance, a creative activity that poses both physical and cognitive demands, has been shown to address motor impairments in people living with PD in a number of small studies [13]. There is evidence that long-term dance practice can modify motor symptom progression [14] and that it can improve balance more effectively than other forms of physical activity [15, 16]. Dance has also been demonstrated to show beneficial effects on gait variables in PD, such as velocity and stride length [17, 18]. More recently, researchers have begun investigating dance's potential effects on non-motor PD symptoms, such as cognitive impairment and depression [19]. Depending on the dance style or technique being practiced, dance classes may train a variety of cognitive skills. For example, dance may improve attention and memory, which are important for learning new dance steps and choreographic sequences. Results thus far have demonstrated that dance can positively impact spatial cognition [20], cognitive switching [21], and mental rotation abilities [22]. In addition to engaging cognitive processes, dance is typically practiced in a social, enjoyable environment, which may reduce isolation and impart psychological benefits [23].

Through the amelioration of motor and non-motor symptoms, QOL may improve in people living with PD. Health-related QOL is defined as the impact an illness and its consequences have on a person as determined by their own perceptions and evaluations [24]. There are a number of mechanisms through which dance may improve QOL including but not limited to improved motor function [24], engagement with music [25], and socialization [26, 27]. Hackney and Bennett concluded in 2014 that more rigorous research is needed to confirm the effects of dance on QOL and to uncover the mechanisms that may be responsible for positive change in this area [25].

Several systematic reviews and meta-analyses on dance and PD have been published over the past decade [13, 28–30], with some focusing specifically on one dance style, such as Argentine tango [31], and others on specific outcomes, such as gait and cognition [32] or non-motor symptoms [19]. Since the most recent comprehensive systematic reviews of the literature, a number of studies investigating novel dance interventions [33–37] and novel outcomes [35, 38–40] have been published. There is thus a need to combine new evidence with previous research to provide a more comprehensive picture of the efficacy of this multifaceted intervention on motor and non-motor symptoms in PD.

The primary aim of this review is to evaluate high-quality evidence in the form of a randomized controlled trial (RCT) design to investigate the efficacy of dance in improving both motor and non-motor symptoms of PD. The secondary aims of this review are to assess the methodological quality of included studies and to inform the direction of future research, thus updating the findings of previous reviews [13, 19, 28–32].

## Methods

### Criteria for considering studies (S1)

**Types of participants.** Participants included people diagnosed with PD, as determined by the authors of included studies. All disease stages, disease durations, and ages were eligible for

inclusion. In order to compare across studies, the disease stage must have been measured and reported using the original or modified Hoehn and Yahr scale (H&Y) [41]; studies that did not report this variable were excluded.

**Types of interventions.**   All interventions must have exclusively used dance as the rehabilitation technique of interest. The definition of dance used was inclusive, including all styles and techniques (i.e., tango, Irish set dancing, ballet, etc.) in all settings (i.e., community centers, rehabilitative centers, etc.).

**Types of comparisons.**   All peer-reviewed randomized controlled trials (RCTs) that compared dance to either no intervention or to an active control, including but not limited to exercise and educational programs, were eligible. RCTs comparing two different dance interventions were also included. Quasi-randomized trials, cohort studies that did not include a control group, and controlled studies that did not implement random treatment allocation methods were not eligible for inclusion in this review.

**Types of outcomes.**   Trials that reported at least one motor outcome (e.g., gait and balance outcomes), one non-motor outcome (e.g., cognitive or mental health related outcomes), or measure of QOL, either self-reported or observed, were included.

## Search method for identifying studies

An electronic systematic search of five databases (Medline, Embase, PsycINFO, Cumulative Index to Nursing and Allied Health Literature [CINAHL], and PubMed) was conducted through week four of March 2020. MeSH terms Parkinson disease, Dance Therapy and Dancing and entry terms Parkinson* and danc* were searched for within articles (see S2 File for full search strategy). No protocol was published or registered prior to conducting the search.

## Selection of studies

From the search results, two review authors (A.M.C. & R.S.) independently screened the abstracts of potentially relevant studies. If the abstract did not provide enough information, the full text was obtained to determine the study's eligibility for inclusion in this review. If the full text was not available, or if trial details were unclear, authors of potentially relevant studies were contacted for additional information. Any disagreements were resolved through discussion between review authors.

## Data extraction

The studies selected for inclusion in this review were then assessed for risk of bias and trial details and data were extracted. The following trial details were recorded for each study: authors, publication year, type of dance, comparison or control group, intervention parameters, the number of participants randomized, the number of participants analyzed, the number of participants who dropped out or were withdrawn, the method of analysis used (i.e., intention-to-treat or per protocol), mean age of participants, and mean H&Y score (see Table 1). Outcomes reported, whether participants were tested ON or OFF medication (or if this was not stated), and a summary of results were also recorded and synthesized qualitatively.

## Assessing risk of bias in included studies

The Cochrane Collaboration risk of bias assessment tool [42] was used to evaluate the methodological quality of all studies included in this systematic review, to assess improvements in trial quality that may have occurred over time, and to provide recommendations for further improvements in future trials. All included studies were assessed for risk of bias in the nine

**Table 1. Characteristics of included studies.**

| Study ID | Dance Style | Control | Intervention Parameters | Randomized (Analyzed) | Dropouts/ Withdrawals | Analysis: Intention to Treat used? | Mean age | Mean H&Y |
|---|---|---|---|---|---|---|---|---|
| Duncan & Earhart [14] | Tango (n = 26) | No intervention (n = 26) | 60 mins, 2x/week, 12 mos. | 62 (52) | 27 | Yes (included participants retained through 3 mos.) | 69.2 | 2.6 |
| Duncan & Earhart [62] | Tango (n = 5) | No intervention (n = 5) | 60 mins, 2x/week, 24 mos. | 10 (10) | 0 | Not stated | 67.8 | 2.4 |
| Foster et al. [63] | Tango (n = 26) | No intervention (n = 26) | 60 mins, 2x/week, 12 mos. | 62 (52) | 27 | Yes (included participants retained through 3 mos.) | 69.2 | 2.3 |
| Hackney et al. (2007) | Tango (n = 9) | Traditional Exercise (n = 10) | 60 mins, 2x/week, 13 weeks (20 sessions) | 19 (19) | 0 | Not stated | 71.1 | 2.3 |
| Hackney & Earhart [17] | Tango (n = 14) Waltz/Foxtrot (n = 17) | No Intervention (n = 17) | 60 mins, 2x/week, 13 weeks (20 sessions) | 58 (48) | 10 | No | 67.0 | 2.2 |
| Hackney & Earhart [24] | Tango (n = 14) Waltz/ Foxtrot (n = 17) | No intervention (n = 17) Tai Chi (n = 13) | 60 mins, 2x/week, 13 weeks (20 sessions) | 75 (61) | 13 | No | 66.6 | 2.1 |
| Hackney & Earhart [18] | Partner (n = 19) Non-Partner Tango (n = 20) | N/A | 60 mins, 2x/week, 10 weeks (20 sessions) | 39 (39) | 12 | Yes | 69.6 | 2.3 |
| Hulbert et al. [39] | Ballroom/Latin American (n = 12) | No intervention (n = 12) | 60 mins, 2x/week, 10 weeks | 27 (24) | 3 | No | 72.6 | 1.9 |
| Kunkel et al. [33] | Ballroom/Latin American (n = 31) | No intervention (n = 15) | 60 mins, 2x/week, 10 weeks | 51 (31) | 5 | No | 70.5 | 2.1 |
| Lee et al. [43] | Turo PD/Qi dance (n = 25) | No intervention (n = 16) | 60 mins, 2x/week, 8 weeks | 32 (32) | 6 | Yes | 65.7 | 1.9 |
| Michels et al. [34] | DT (n = 9) | Support group (n = 4) | 60 mins, 1x/week, 10 weeks | 13 (13) | 0 | Not stated | 69.2 | 2.3 |
| Rocha et al. [36] | Tango (n = 8) Mixed genre (n = 10) | N/A | 60 mins, 1x/week, 8 weeks + 40 min home prog. | 21 (18) | 8 | Yes | 71.6 | 2.5 |
| Rios Romenets et al. [67] | Tango (n = 18) | Self-directed exercise (n = 15) | 60 mins, 2x/week, 12 weeks | 33 (33) | 4 | Yes | 63.8 | 1.9 |
| Shanahan et al. [64] | Irish set dancing (n = 20) | No intervention (n = 21) | 90 mins, 1x/week, 10 weeks + 20-min home prog. | 90 (41) | 28 | No | 69.0 | 1–2.5 |
| Solla et al. [65] | Sardinian folk dancing (n = 10) | No intervention (n = 9) | 90 mins, 2x/week, 12 weeks | 20 (19) | 1 | Not stated | 67.5 | 2.2 |
| Volpe et al. [66] | Irish set dancing (n = 12) | Physiotherapy (n = 12) | 90 mins, 1x/week, 6 mos. + 60 min home prog. | 24 (24) | 0 | Not stated | 63.3 | 2.2 |

categories that are considered standard features of interest in parallel group trials: random sequence generation (selection bias), allocation concealment (selection bias), blinding of participants and personnel (performance bias), blinding of outcome assessments (detection bias), incomplete outcome data (attrition bias), participant similarity at baseline, intention to treat (ITT) analysis, eligibility criteria, consistency of co-interventions, and comparability between trial arms [42]. Given that it is impossible to blind participants and those delivering the intervention (i.e., dance instructors) in such trials, all studies were deemed to have a high risk of

performance bias. Despite the inevitability of this result, it was included in the risk of bias assessment to provide a clear picture of the overall level of bias. Each study was classified as having a low, unclear, or high risk of bias in each of the nine categories, and justification for each decision was provided (see S3 File).

## Synthesis and analysis of data

The results of all studies were synthesized qualitatively, and when appropriate meta-analyses were conducted using the Cochrane Collaboration Review Manager Software (version 5.3) to compare the efficacy of dance interventions to active controls (e.g., physiotherapy, educational programs, etc.) or usual care (i.e., no intervention). All outcome variables analyzed, which included measurements of motor symptoms, non-motor symptoms, and QOL, were continuous data. Pooled effect estimates were calculated from the mean change in scores from baseline to post-test, their standard deviations (SDs), and the number of participants analyzed. If standard errors of the mean were reported in lieu of SDs in publications, SDs were calculated for meta-analysis purposes. Only one trial reported change SDs [43]. For all other studies, change SDs were imputed with the correlation coefficient set at 0.5, a value reported as conservative [44]. In trials where two types of dance were compared to another active control or no intervention [17, 24], the means and SDs of the two dance groups' change scores were pooled. This approach was taken because the aim of the meta-analysis was to compare dancing to either no intervention or an active control rather than to compare different styles of dance [29]. RCTs that solely compared two types of dance [18, 36] were not eligible for inclusion in the meta-analysis. If data were only reported in graph form in a publication, authors were contacted via email and asked to provide means and standard deviations for all groups at all time points.

## Results

### Outcomes of literature search and characteristics of included studies

**Studies included and excluded.** Thirty-four trials that evaluated the efficacy of dance for people with PD were identified from this search (see Fig 1). Sixteen met the outlined eligibility criteria and 18 were excluded. Reasons for excluding full-text articles reviewed included: used a "quasi" method of randomization (n = 1) [22], convenience sampling (n = 2) [45, 46], and disease stage of participants not reported (n = 2) [47, 48]. Thirteen trials presented in conference abstracts were also assessed for eligibility by contacting authors for more details. All 13 were excluded with the reasons being participants were partially randomized (n = 1) [49], participants were not randomized (n = 1) [50], or additional information about the trial could not be accessed to assess eligibility (n = 11) [51–61].

**Participants.** The number of participants randomized in each individual trial ranged from 10 to 90 and the number of participants analyzed ranged from 10 to 61. Thus, 636 participants were randomized and 516 were analyzed in total, which makes for an average trial size of 40 participants with an average of 32 included in the analysis. The average age of participants was 68.4 and the average H&Y stage was 2.2, which indicates mild to moderate disease severity.

**Characteristics of included studies.** Sixteen trials were included in this review (see Table 1 for detailed characteristics of each trial). Thirteen were parallel, between-subject two-arm trials, one was a three-arm trial, one a four-arm trial, and one a partial crossover design. Seven of the two-arm trials compared a dance intervention to no intervention [14, 33, 39, 62–65], one compared dance to standard physiotherapy exercises [66], one compared dance to self-directed exercise [16], one compared dance to traditional exercise [15], one compared

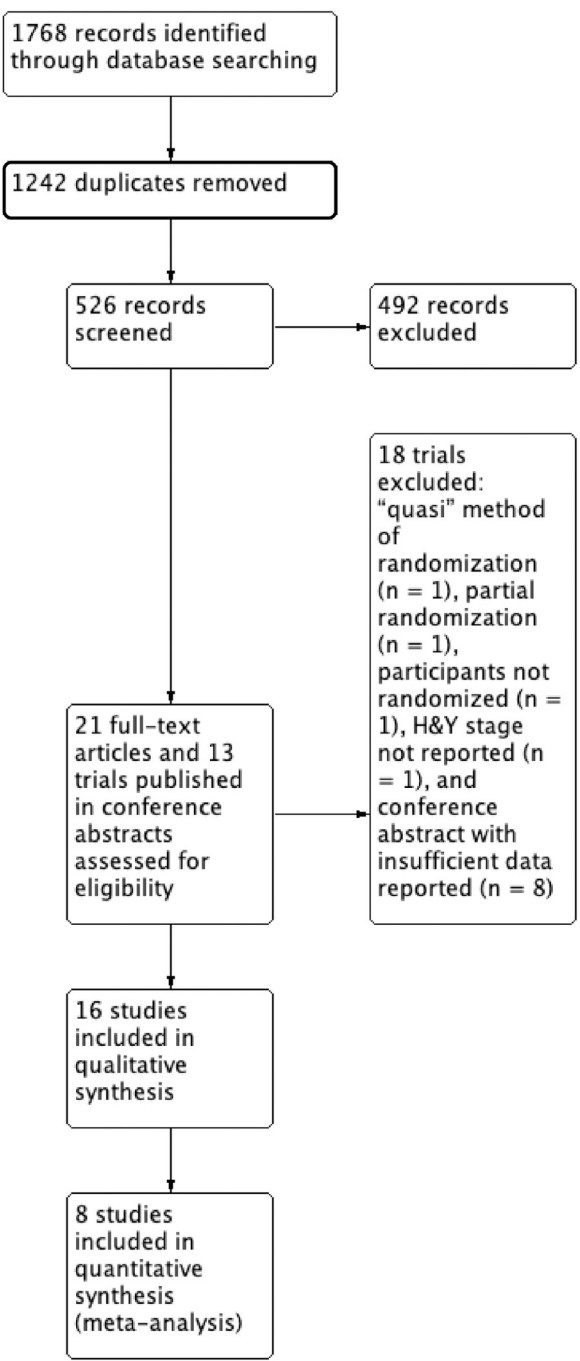

**Fig 1. PRISMA flow diagram.**

dance to support group sessions [34], and two compared two types of dance, including partnered and non-partnered tango [18] and tango and mixed genre [36]. The three-arm trial compared two different types of dance (i.e., tango and waltz/foxtrot) to no intervention [17] and the four-arm trial compared two different types of dance (i.e., tango and waltz/foxtrot) to Tai Chi and no intervention [24]. The trial that used a partial crossover design compared Turo PD, a Qigong dance hybrid, to no intervention [43].

Eleven of the 16 interventions included 60-minute dance classes that met two times per week, with interventions lasting eight weeks to two years in duration [14–18, 24, 33, 39, 43, 62, 63]. One intervention included a 90-minute dance class that met two times per week for 12 weeks [65]. One intervention included a once-weekly 60-minute dance class with a 40-minute home program for eight weeks [36], and two included 90-minute dance classes with 60-minute home programs with the interventions lasting 10 weeks and six months, respectively [64, 66]. One included a 60-minute dance therapy session practiced once per week for 10 weeks [34]. Thus, the amount of time spent dancing ranged from 60 to 180 minutes per week.

With regard to dance style, nine of the 16 RCTs (56%) evaluated the effects of tango [14–18, 24, 36, 62, 63]. Four studies evaluated the effects of different types of ballroom and/or Latin dance styles other than tango [17, 24, 33, 39], two evaluated Irish set dancing [64, 66], one evaluated a Sardinian folk dance called Ballu Sardu [65], and one evaluated a mixed dance genre that incorporated tap dancing, creative dance, and Irish set dancing [36]. Only one study evaluated a non-Western style of dance, Turo PD/Qi dance [43]. Only one trial evaluated a dance therapy program led by a dance therapist [34]. The other interventions were led by a professional ballroom dance instructor and personal trainer [15, 17, 18, 24], dancers with extensive performance experience [62], professional instructors without PD expertise [16], instructors supervised by physical therapists [33, 34, 36, 63, 65], and set dancing teachers who were clinicians or had experience working with clinical groups [64]. Three studies did not describe the qualifications of the instructors [14, 43, 66].

**Risk of bias of included studies.** All studies included in this systematic review were assessed for risk of bias in nine categories: random sequence generation (selection bias), allocation concealment (selection bias), blinding of participants and personnel (performance bias), blinding of outcome assessment (detection bias), incomplete outcome data (attrition bias), participant similarity at baseline, ITT analysis, eligibility criteria, consistency of co-interventions, and comparability between trial arms. The results are presented in Figs 2 and 3. Fig 2 presents the authors' judgements (low, unclear, or high risk) as percentages across all included studies, and Fig 3 presents the specific judgement ratings (low, unclear, or high) for each risk of bias category for each study. These categories were determined using the Cochrane Risk of Bias Tool [42]. Justification for each decision is outlined in the Risk of Bias Tables (S3 File).

Because it is not possible to control for performance bias in the context of a dance intervention due to the inability to blind participants and personnel to the intervention being delivered, controlling for selection and detection bias is particularly important. With regard to selection bias, less than half of the studies used a low risk randomization procedure (see Fig 2) and only two described methods used to conceal allocation (see Fig 3) [64, 66]. All but three [16, 24, 39] used blinded assessors to mitigate the risk of detection bias. Several trials also attempted to reduce the risk for performance bias by blinding participants to the study hypotheses [16–18, 36]. In Rocha et al.'s [36] trial, which compared the effects of tango to a mixed dance program, all participants, dance teachers, and assistants were blinded to the study aims. In Volpe et al.'s [66] trial evaluating Irish set dancing, all staff involved in usual care were blinded to the study aims and hypotheses and were subsequently tested at the end of the trial to see if the blinding protocol was effective. Approximately 30% of staff members correctly guessed the group assignment [66].

With regard to participant characteristics, inclusion and exclusion criteria were clearly described in all trials, and three of the 16 trials [16, 33, 34] reported statistically significant differences between groups at baseline. These three trials accounted for 92 of the 516 participants analyzed (18%). The baseline differences reported in these trials included a higher fall risk and a greater propensity to exercise in the self-directed exercise group compared to the tango group [16], older age and higher mean MDS-UPDRS III scores at baseline in the support

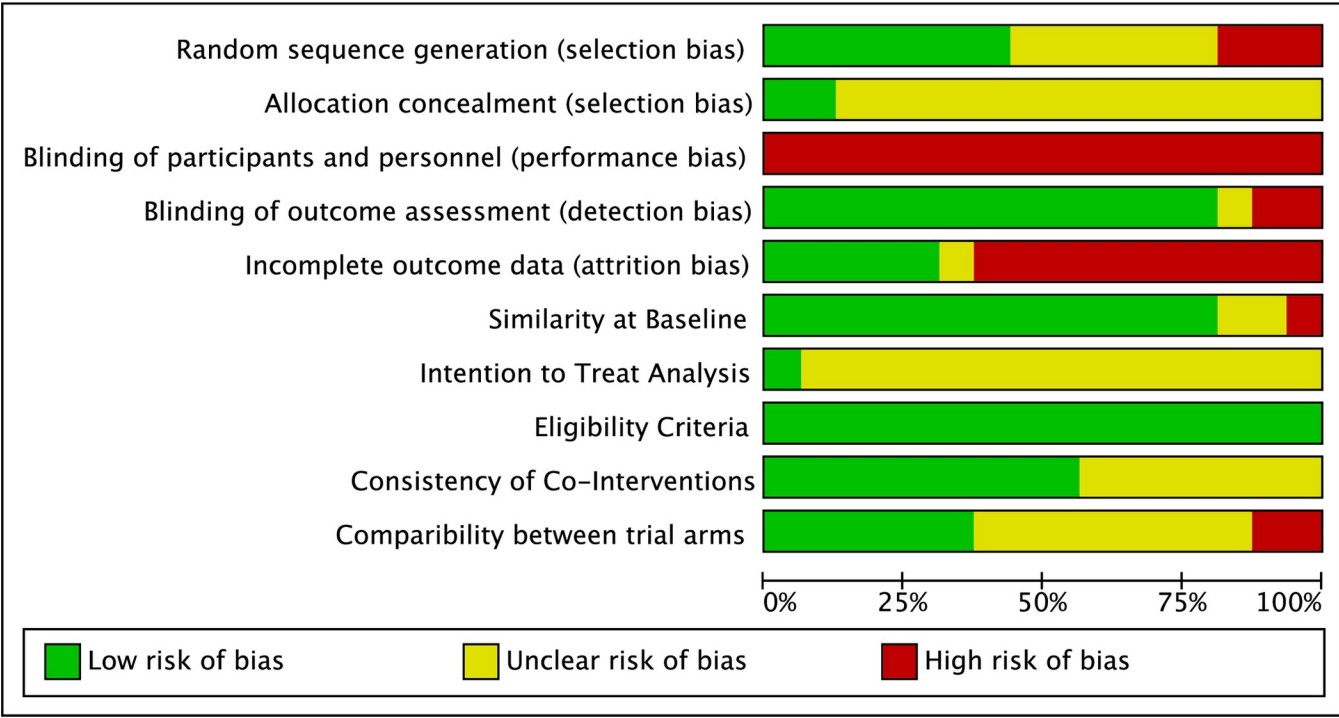

**Fig 2. Risk of bias graph.** Review authors' judgements concerning each risk of bias item presented as percentages across all included studies.

group as compared to the dance therapy group [34], and a trend toward longer time since diagnosis in the no intervention control group compared to the ballroom/Latin American dance group [33].

The consistency and reporting of co-interventions varied considerably across trials. Seven trials included in this review controlled for medication during the course of the interventions through monitoring [16, 17, 24, 34, 43, 64, 65], with four excluding participants who experienced medication changes [17, 24, 34, 64]. Six stated that participants continued with usual care while participating in the interventions [17, 18, 24, 33, 39, 65]. Ten trials controlled for medication-related fluctuations in performance during assessment sessions by either testing participants "OFF" medication [14, 62, 63] or during an "ON" state at a standardized time of day [15, 17, 24, 28, 33, 36, 39]. Three did not report the medication state of participants during assessments [34, 43, 64]. In two trials, participants were tested "ON" medication, but it is not described if this occurs at a particular time of day or period of the medication cycle [16, 65]. Volpe et al. [66] reported that assessments did not always occur at the peak dose in medication cycles despite always taking place at a standardized time of day.

Nine trials instructed participants to continue with their regular exercise routines or "usual activities" outside of the intervention; however, the level of exercise or activity engaged in is not recorded or quantified [15, 17, 18, 34, 36, 38, 55, 64, 65]. The control group in Rios Romenets et al.'s [16] study continued with usual care and participants were given the option of either continuing with their regular exercise regime if it was considered 'intensive' by the research team or being prescribed a self-directed exercise program if they were not already engaged in intensive regular exercise. It was not described how the research team determined if participants' exercise schedules were intensive nor was it described how activity levels or adherence to the self-directed program were monitored [16]. Only one trial reported using an exercise diary to monitor compliance with a home exercise program that was a part of the

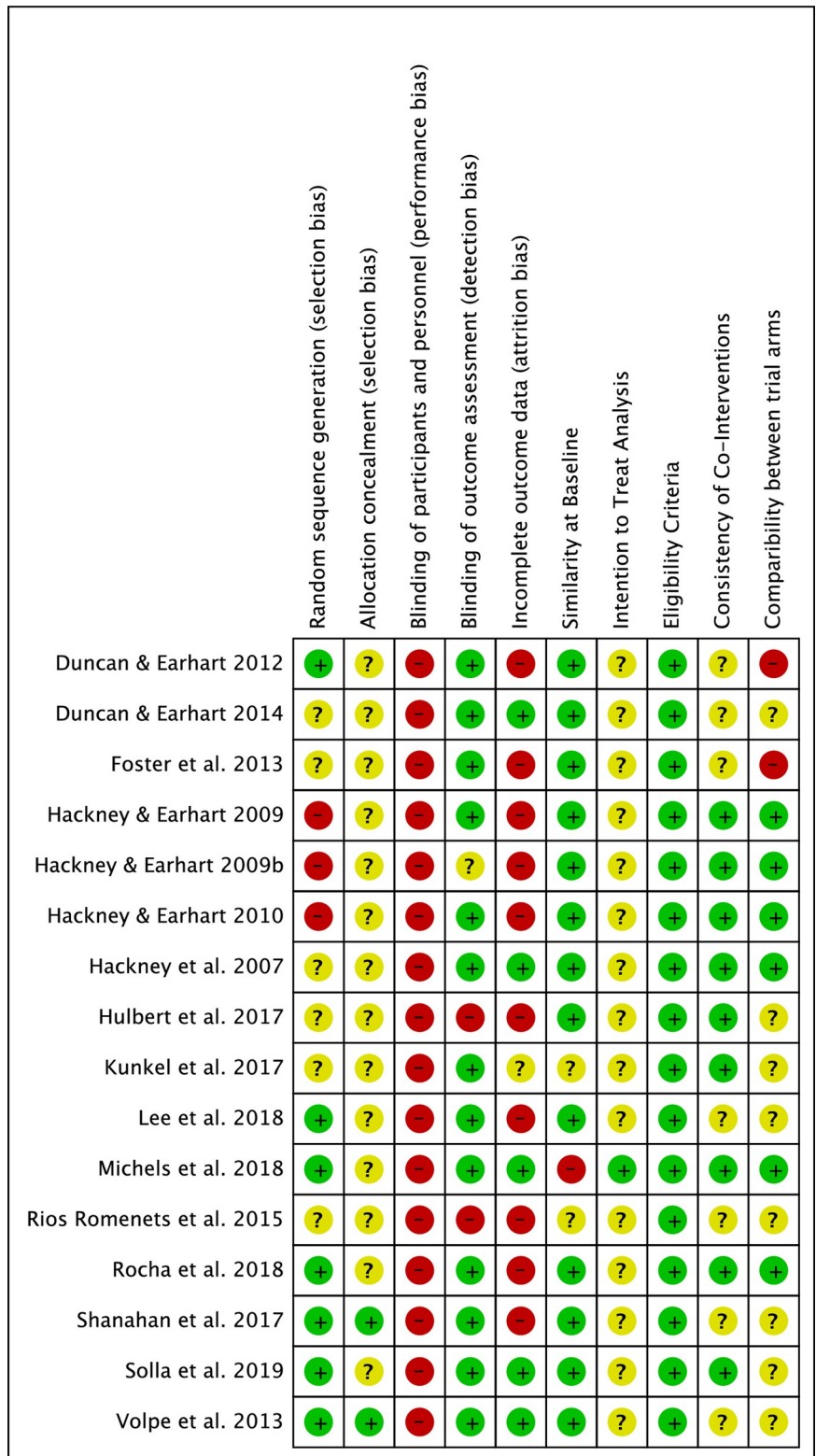

**Fig 3. Risk of bias summary.** Review authors' judgements concerning each risk of bias item for each included study.

intervention; again, the diary was not used to quantify participants' exercise levels outside of the dance intervention [64]. Six trials were rated as having a low risk of bias regarding comparability of trial arms as they included active control groups with equal contact time for at least two groups included in the trials [15, 17, 18, 24, 34, 36]. Volpe et al. [66] was rated as having an unclear risk of bias in this category despite having an active control because the control group received individual physiotherapy sessions, while the Irish set dancing was taught in a group setting.

With regard to attrition bias, studies were classified as having a low risk of bias if the number of dropouts or withdrawals was less than 10% and a high risk if greater than 10% [5]. Four of the 16 studies reported no dropouts or withdrawals [15, 34, 62, 66], and one had no dropouts but withdrew a participant from the analysis due to severe dyskinesia and freezing [65]. The remaining 11 all had dropouts greater than 10%. All studies with dropouts and withdrawals provided details if possible explaining why participants discontinued the interventions or why they were excluded from the analyses. Reasons for discontinuing the trials due to issues related to the dance interventions included lack of interest [18], too fatiguing [18, 64], disliked dance [33], did not meet needs [36], did not like the program [64], and too much to handle [14, 63].

With regard to method of analysis, all but one study was classified as having an unclear risk of bias. Reasons for having an unclear risk of bias included the method of analysis was not explicitly described [15, 33, 62, 65, 66], ITT analysis was reported as used but not all participants randomized were analyzed [14, 36, 63], per protocol analysis was used [17, 24, 39, 64], or ITT was used with dropouts greater than 10% [16, 18, 43]. Only one intervention was classified as having a low risk of bias because a comparison of outcomes was not carried out due to the study being underpowered [34].

Attempts to mitigate the risk of publication bias in this review were made by carrying out a systematic search not limited by outcome and attempting to include trials presented in conference meeting abstracts [42]. Of the meeting abstracts considered, two were not eligible for inclusion and the data of the remaining 11 were not accessible because authors either could not be reached or did not reply to the request for additional information. The lack of grey literature is a limitation of this review as it reflects an incomplete retrieval of identified research. Of studies included, the majority reported one or more null/neutral result [14–18, 24, 34, 36, 39, 43, 62, 65, 66] or no significant changes whatsoever [33, 64].

## Qualitative synthesis

All 16 trials were included in the qualitative synthesis of this review, which describes the effects of each intervention on motor impairments, non-motor symptoms, and QOL.

**Motor symptom severity.** The motor component of the Movement Disorder Society United Parkinson's Disease Rating Scale Part III (MDS-UPDRS-III) was reported in 11 studies [14–17, 34, 36, 43, 62, 64–66].

In the six studies that compared dance to no intervention, four showed improvement in motor symptom severity after practicing tango, Qi dance, and Sardinian folk dance in comparison to controls [14, 43, 62, 65]. The other two saw controls worsen in disease severity, and no changes in the tango or Irish set dance groups [17, 64].

Among the three studies that compared dance to physiotherapy or exercise, one saw similar improvements in motor symptom severity in both the tango and exercise groups [15], one saw no changes in either the tango or self-directed exercise groups [16], and one saw improvements in both the Irish set dance group and the physiotherapy group, with better results in the dance group [66]. One study that compared the effects of dance therapy to attending support

group sessions was not powered to assess significant differences; however, the authors reported greater positive change in the dance therapy group (-4.12) compared to the support group (-1.75) [34].

Two studies compared the effects of two styles of dance on motor symptom severity; one showed no group differences in MDS-UPDRS III in comparing tango and mixed dance programs [36], and the other found no change in either tango or waltz/foxtrot groups but a worsening in the no intervention control group [17].

**Balance.** Thirteen of the 16 studies measured changes in balance as part of their analyses, with four using the Mini-BESTest [14, 16, 62, 64] and nine using the Berg Balance Scale (BBS) [15, 17, 18, 33, 34, 36, 43, 65, 66].

Three studies compared dance to no intervention using the Mini-BESTest, and two reported improvements in balance among tango groups compared to no intervention [14, 62]. The third study reported no differences in balance post-intervention between the Irish set dancing group and no intervention controls [64]. Four studies used the BBS to compare dance to no intervention; two showed no improvement in the dance groups practicing ballroom/ Latin American dance or Qi dance in comparison to usual care [33, 43] while the other two showed improvement in Sardinian folk and tango and waltz/foxtrot dancers compared to no intervention controls [17, 65].

The one study that compared tango to an active control using the Mini-BESTest found improvement in the tango group compared to the self-directed exercise group [16, 67]. Among the three studies that used the BBS to compare dance to an active control, one reported the tango group improved in balance while the exercise group did not [15], one saw a trend toward improvements after Irish set dancing and physiotherapy but no significant differences between groups [66], and one saw no improvement after dance therapy in comparison to attending a support group [34].

Both studies that compared two different dance interventions used the BBS to assess changes in balance. Hackney and Earhart [18] found improvements among both partnered and non-partnered tango groups, but there were no differences between groups. Rocha et al. [36] showed improvement in the tango group but not in the mixed dance group, but there were no significant differences between groups. Hackney and Earhart [18] additionally used the Tandem Stance (TS) and One Leg Stance (OLS) tests to compare the effects of partnered and non-partnered tango and found significant positive changes in both groups after 20 lessons, with TS maintaining significance at one-month follow-up.

Only one study evaluated subjective changes in balance confidence. Kunkel et al. [33] used the Activities Specific Balance Confidence Scale to compare ballroom/Latin American dance to a no intervention control reporting no changes in either group.

**Gait.** Seven studies measured changes in gait. Among the studies that compared dance to a no intervention control, Duncan and Earhart [14] found improvement in comfortable forward and dual task walking velocities measured using GAITRite following six and 12 months of tango practice in comparison to no intervention, while Duncan and Earhart [62] found no interactions or effects after 24 months of tango. Hackney and Earhart [17] reported improvement in backwards stride length after 13 weeks of either adapted tango or waltz/foxtrot compared to a no intervention control and worsening in forward and backwards single support percent in controls compared to both dance groups. Using a wearable gait analysis system, Solla and colleagues [65] measured changes in several gait variables, including walking speed, cadence, stride length, number of straight walks, straight walking time, and gait fatigue index (GFI). The Sardinian folk dance group improved in stride length, walking speed, and straight walk variables in comparison to the no intervention control. The controls also experienced a significant worsening in GFI, which was based on a decrease in gait speed, while the dance

group showed a trend towards improvement [65]. Only one study compared changes in gait following a dance intervention to an active control and no changes were found in either the tango or exercise groups [15].

Two studies compared the effects of different dance programs on gait [18, 36]. In 2010, Hackney and Earhart reported improvement in comfortable and fast as possible walking velocities, cadence, and double support percent after 10 weeks of both partnered and non-partnered tango; these effects were maintained at a one-month follow up assessment [18]. In 2018, Rocha et al.'s study compared the effects of two different types of dance (tango and mixed dance) on gait and found no effects [36].

**Freezing of gait.** Seven studies evaluated changes in freezing of gait using the Freezing of Gait Questionnaire (FOG). Two studies compared tango to no intervention [14, 62]. Duncan and Earhart [14] found a group by time interaction for FOG, with no intervention controls reporting more freezing after 12 months compared to baseline, while Duncan and Earhart [62] found no effects of dance on FOG nor differences between groups over time.

Three studies compared the effects of dance on FOG to an active control. Volpe et al. [66] reported improvement following six months of Irish set dancing compared to physiotherapy, which showed no improvement. Hackney et al. [15] and Rios Romenets et al. [16] found no significant changes in tango groups in comparison to traditional group exercise and self-directed exercise, respectively.

Among the studies that compared two types of dance, one reported improvement in FOG after participation in eight weeks of mixed genre but not tango classes [36], while the other reported no significant differences in tango, waltz/foxtrot, and no intervention control groups after 13 weeks [17].

**Endurance.** Seven studies evaluated endurance and aerobic capacity using the Six Minute Walk Test (6MWT), which measures the distance a person is able to walk in six minutes. Five of these trials compared dance to no intervention [14, 33, 62, 64, 65]. Shanahan et al. [64] reported no changes in endurance following an eight-week Irish set dancing program and Kunkel et al. [33] showed a trend toward improvement after 12-weeks of ballroom/Latin American dance. Duncan and Earhart [14] and [62] similarly showed no change in endurance after 12 and 24 months of tango, respectively, yet they reported a worsening in the no intervention control groups [14, 62]. Conversely, Solla and colleagues [65] found an increase in endurance after 12 weeks of Sardinian folk dance compared to no intervention controls.

Two studies compared the effects of two different types of dance on the 6MWT, with one noting improvements in both tango and waltz/foxtrot groups compared to no intervention controls after 20 sessions [17], and the other finding a trend toward improvement at post-testing and significant improvement at follow up after both 20 sessions of partnered and non-partnered tango [18].

**Functional mobility.** Ten trials evaluated functional mobility using the Timed Up and Go test (TUG). Two evaluated the effects of tango in comparison to no intervention finding no effects [17, 62], while two others reported improvements in ballroom/Latin American and Sardinian folk dance groups in comparison to no intervention [33, 65].

Four compared dance to an active control [15, 16, 34, 66]. One showed improvement in favor of tango compared to self-directed exercise [16] and another showed improvement in favor of Irish set dance compared to physiotherapy [66]. There were no improvements following a dance therapy intervention compared to a support group [34] nor a tango intervention compared to exercise [15].

Two compared the effects of two different types of dance on functional mobility, with Hackney & Earhart [18] finding no effects as a result of either partnered or non-partnered

dance, and Rocha et al. [36] reporting improvement in the tango group but not in the mixed dance group.

The Dual-Task TUG was reported in two studies. Duncan and Earhart [62] found an effect of time, with the tango group improving and the no intervention controls worsening after 12 months; however, there were no differences between groups. Rios Romenets et al. [16] reported improvements in Dual-Task TUG scores in the tango group in comparison to the self-directed exercise group after 12 weeks.

**Coordination while turning.** One study evaluated the effects of whole-body coordination while turning in four conditions (predicted preferred, predicted un-preferred, unpredicted preferred, and unpredicted un-preferred) [39]. It was reported that those who participated in a 10-week ballroom/Latin dance program were able to better coordinate axial and perpendicular body segments and turned more 'en bloc' (i.e., with tighter coupling of body segments) in comparison to no intervention controls. Hulbert et al. [39] and Kunkel et al. [33] both measured turning ability using the Standing Start 180; no differences were found in the dance or no intervention groups.

**Upper extremity function.** Two studies reported measures of upper extremity function and manual dexterity. One used the Perdue Pegboard Test to compare tango to self-directed exercise [16] and the other used the Nine Hole Peg Test to compare tango to no intervention [14], with only the latter finding tango to lead to improvement in upper extremity and hand function in comparison to no intervention. One study measured upper body flexibility using the back scratch test and found improvements after 12 weeks of Sardinian folk dancing compared to no intervention [65].

**Lower extremity function.** One study measured lower extremity function using the Five Times Sit-to-Stand test (FTSST) and the Sit and Reach Test (SRT) [65]. Solla and colleagues found improvement in lower limb strength measured using the FTSST in the Sardinian folk dance group while the no intervention control group worsened. No differences in lower body joint mobility measured using the SRT were found between the dance and no intervention groups [65].

**Posture.** One study measured changes in posture following ballroom/Latin American dance practice using the spinal mouse, a device that assesses curvatures of the spinal column, and reported no effects [33].

**Falls.** Only one study reported an outcome measuring fall frequency during the intervention. Rios Romenets et al. [16] used the Falls Questionnaire (Canadian Community Health Survey), adapted to focus on three months (i.e., the length of the dance intervention), and found no differences in fall frequency following 24 partnered tango classes in comparison to self-directed exercise.

**Cognitive function.** Three studies included in this review measured cognitive function using the Montreal Cognitive Assessment (MoCa), with two reporting improvements following a dance intervention. Rios Romenets et al. [16] found an improvement that approached, but did not reach significance, in the tango group in comparison to the controls who practiced self-directed exercise. A significant improvement was found after the exclusion of protocol violations [16]. Solla et al. [65] saw improvements following 12 weeks of Sardinian folk dancing in comparison to no intervention, with controls demonstrating a slight, non-significant worsening. Michels et al. [34] found no changes after participants engaged in dance therapy or support groups.

**Mental health.** Four studies [16, 34, 43, 65] measured the effects of dance on symptoms of depression using the Beck Depression Inventory and two measured the effects of dance on symptoms of apathy using the Starkstein Apathy Scale [16, 65]. Only Solla and colleagues [65] found significant improvements in symptoms of depression and apathy following 12 weeks of

Sardinian folk dance in comparison to usual care. No changes were seen in symptoms of depression after eight weeks of Qi dance [43] or eight weeks of dance therapy [34]. Rios Rome-nets and colleagues [16] similarly saw no improvements in symptoms of depression or apathy after 12 weeks of tango.

**Fatigue.**    Three studies measured changes in fatigue [16, 34, 65]. Two used the Krupp Fatigue Severity Scale, with one reporting borderline significant improvements in the tango group compared to controls who practiced self-directed exercise and significant improvement when protocol violations were excluded [16]. The other found no change following a dance therapy intervention compared to a support group [34]. One used the Parkinson's Disease Fatigue Scale (PFS-16) and reported no difference in perceived fatigue between the Sardinian folk dance and no intervention control groups [65].

**Quality of life (QOL).**    Seven studies reported data measuring QOL using the Parkinson's Disease Questionnaire (PDQ-39) [16, 24, 33, 34, 36, 64, 66] and one used the Parkinson's Disease Quality of Life Questionnaire [43]. Only two studies reported significant improvements in the dance groups compared to controls [24, 43]. In Hackney & Earhart's [24] study, participants who completed 20 adapted tango sessions improved in the PDQ-39 Summary Index and in Mobility and Social Support sub-scores in comparison to those who were assigned to waltz/foxtrot, Tai Chi, or no intervention. Lee et al. [43] reported improvements in the PDQL total and Systemic Symptoms and Social Functioning sub-scores after 16 Qi dance sessions compared to no intervention.

**Experiences of daily living.**    Four studies reported data on the non-motor experiences of daily living using MDS-UPDRS subscale I, three of which compared dance (i.e., tango and Qi dance) to no intervention [14, 43, 62] and one that compared dance therapy to a support group [34]. Five reported data on the motor experiences of daily living using MDS-UPDRS subscale II, with three comparing dance to no intervention [14, 43, 62], one comparing dance therapy to a support group [34], and one comparing tango and mixed dance [36].

Duncan and Earhart [14] found no differences between intervention arms (tango vs. no intervention) in either MDS-UPDRS subscales I or II after 12 months [14]. However, in 2014, Duncan & Earhart found that 12 and 24 months of tango improved non-motor experiences of daily living (MDS-UPDRS I) and motor experiences of daily living (MDS-UPDRS II) in comparison to no intervention [62]. Lee et al. [43] found an improvement in the UPDRS ADL sub-scale but not Mentation and Mood subscales after eight weeks of Qi dance compared to no intervention. No differences were found in the studies that compared tango and mixed dance [36] or evaluated a dance therapy program in comparison to a support group [34].

**Participation.**    Foster et al. [63] used the Activity Card Sort, which measures changes in activity participation but does not focus specifically on the experience of PD. They found total current participation increased in the tango group compared to no intervention controls, with total activity retention increasing from 77% to 90% in the tango group.

**Clinical global impression of change.**    One study measured changes in Clinical Global Impression of Change from the perspective of the participant and examiner, reporting significant changes in favor of the tango group in comparison to the self-directed exercise group from the examiner's perspective only [16]. The rater was not blinded to the interventions in this study, and it is acknowledged by the authors that subjective factors may have influenced this outcome.

## Meta-analysis

To combine and further analyze the intervention effects of dance on PD, a meta-analysis was conducted on a subset of selected trials. Due to the variety of intervention parameters in

included studies, only trials with the most common intervention length (8–12 weeks) and dosage (60–90 minutes, 2x per week) were included in the meta-analysis to control for clinical diversity. Data gathered at the 12-week time point in Duncan and Earhart's [14] 12-month intervention were included. Unique outcome measures that were only used in one trial, such as the Activity Card Sort [63] and the Spinal Mouse [33], could not be analyzed statistically. Thus, 10 outcome measures and eight trials were included in the meta-analysis, six of which compared dance to no intervention [14, 17, 24, 33, 43, 65] and three of which compared dance to an active control [15, 16, 24].

Heterogeneity was measured using the $I^2$ statistic and was assessed according to Cochrane standards (i.e., 0–40% = potentially important heterogeneity; 30–60% = moderate heterogeneity; 50–90% = substantial heterogeneity; 75–100% = considerable heterogeneity) [42]. Since clinical and methodological diversity were controlled for by only including studies with similar intervention parameters, and all studies included participants with mild to moderate PD of a similar age, a fixed effects inverse variance model was used, and heterogeneity values were reported but were not used to exclude trials from the meta-analysis. For all outcomes apart from those evaluating balance, the mean difference (MD) with 95% confidence intervals was calculated using a fixed effects inverse variance model. To analyze balance, which was measured in trials using two different scales (BBS and Mini-BESTest), the standardized mean difference (SMD) with 95% confidence intervals was calculated using a fixed effects inverse variance model. Tests for funnel plot asymmetry could not be carried out to detect reporting biases because none of the meta-analyses for any outcome included more than 10 studies [42].

**Dance vs. no intervention.**   To evaluate the effects of dance in comparison to no intervention on motor impairments, QOL, and symptoms of depression, meta-analyses were conducted on the following outcomes: MDS-UPDRS III, balance, forward and backward gait velocity, stride length, FOG-Q, 6MWT, TUG, PDQ-39, and BDI-II (Fig 4).

**MDS-UPRDS III.**   The effects of dance on motor symptom severity (MDS-UPDRS III) (Fig 4.1) were assessed by combining data from four studies (n = 160) comparing dance to no intervention. The results favored dance (-2.31 points, CI -3.57 to -1.04, p = 0.0004). There was potentially important heterogeneity (P = 0.24, $I^2$ = 29%) despite controlling for variation in clinical diversity [14, 17, 43, 65].

**Balance.**   The effects of dance on balance (Fig 4.2) were investigated by combining data from five studies (n = 206) contrasting dance to no intervention. The results favored dance (0.50 points, CI 0.21 to 0.79, p = 0.0007); however, there was moderate heterogeneity (P = 0.14, $I^2$ = 42%) despite controlling for clinical diversity and the results showing consistent directions of effects [14, 18, 33, 43, 65].

**Gait variables.**   The effects of dance on forward gait velocity (Fig 4.3) were assessed by combining data from three studies (n = 119) [14, 17, 65] and the effects of dance on backward gait velocity (Fig 4.4) were assessed by combining data from two studies (n = 98) [14, 17]. No significant effect was found for forward gait velocity (0.04 m/s, CI -0.06 to 0.13, p = 0.44), which did not present evidence of heterogeneity (P = 0.71, $I^2$ = 0%). The results for backwards gait velocity were also not significant (0.07 m/s, CI -0.06 to 0.20, p = 0.30) and did not present evidence of heterogeneity (P = 0.47, $I^2$ = 0%). The effects of dance on forward stride length (m) (Fig 4.5) were assessed by combining data from two studies (n = 67) [17, 65]. No significant effect was found for stride length (0.08 m, CI -0.03 to 0.18, p = 0.17) and there was no evidence of heterogeneity (P = 0.91, $I^2$ = 0%).

**FOG-Q.**   The effects of dance on freezing of gait (FOG-Q) (Fig 4.6) were assessed by combining data from two studies (n = 100) contrasting dance to no intervention. The results were not significant (-1.94 points, CI -4.33 to 0.46, p = 0.11) and there was no evidence of heterogeneity (P = 0.89, $I^2$ = 0%) [14, 17].

### 4.1 MDS-UPDRS III

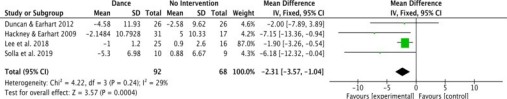

### 4.2 Balance

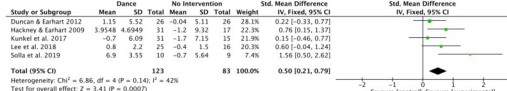

### 4.3 Forward Gait Velocity (m/s)

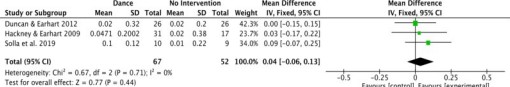

### 4.4 Backward Gait Velocity (m/s)

### 4.5 Stride Length (m)

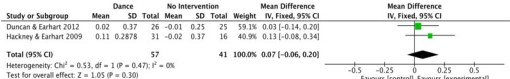

### 4.6 FOG-Q

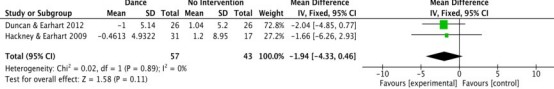

### 4.7 6MWT

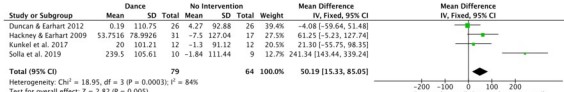

### 4.8 TUG (sec)

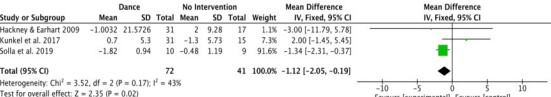

### 4.9 PDQ-39

### 4.10 BDI-II

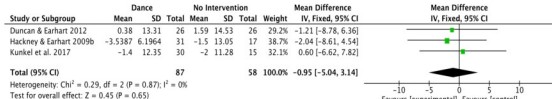

**Fig 4. Meta-analysis results of dance vs. no intervention.**

**6MWT.** The effects of dance on the 6MWT (Fig 4.7) were analyzed by combining four studies (n = 143) comparing dance to no intervention [14, 17, 33, 65]. The results favored dance (50.19 m, CI 15.33 to 85.05, p = 0.005), yet there was substantial heterogeneity (P = 0.0003, $I^2$ = 84%). The heterogeneity seen here is due to the presence of an outlying study [65]. Given the small number of studies included in the meta-analysis, and the fact that clinical diversity was controlled for, it was not seen as appropriate to exclude the study; however, a sensitivity analysis revealed that removing the outlying study investigating Sardinian folk dance resulted in virtually no evidence of heterogeneity (P = 0.34, I = 8%) and no significant effects (22.44 m, CI -14.87 to 59.74, p = 0.24).

**TUG.** The effects of dance on functional mobility (TUG) (Fig 4.8) were analyzed by combining three studies (n = 113) comparing dance to no intervention [17, 33, 65]. The results favored dance (-1.12 seconds, CI -2.05 to -0.19, p = 0.02); however, there was moderate heterogeneity (P = 0.17, $I^2$ = 43%) likely due to different directions of effects.

**PDQ-39.** The effects of dance on health-related QOL using the PDQ-39 (Fig 4.9) were analyzed by combining three studies (n = 145) [14, 33, 24]. The result was not significant (-0.95 points, CI -5.04 to 3.14, p = 0.65) with no evidence of heterogeneity (P = 0.87, $I^2$ = 0%).

**BDI-II.** Only one non-motor outcome was eligible for inclusion in the meta-analysis. The effects of dance on symptoms of depression measured using the BDI-II (Fig 4.10) were analyzed by combining two studies (n = 60) [43, 65]. The result was significant (-5.06 points, CI -7.74 to -2.37, p = 0.0002) with moderate heterogeneity (P = 0.15, $I^2$ = 53%).

**Dance vs. active control.** To evaluate the effects of dance on motor symptoms and QOL in comparison to an active control, meta-analyses were conducted on the following outcomes: MDS-UPDRS III, balance (Mini-BESTest and BBS scores included), FOG-Q, TUG, and PDQ-39 (Fig 5).

**MDS-UPDRS III.** The effects of dance on motor-symptom severity (measured using the MDS-UPDRS III) (Fig 5.1) were analyzed by combining two studies (n = 52) comparing dance to exercise [15, 16]. There were no significant effects (-0.40 points, CI -3.61 to 2.81, p = 0.81) and there was no heterogeneity (P = 1.0, $I^2$ = 0%).

**Balance.** The effects of dance on balance (Fig 5.2) were analyzed by combining two studies (n = 52) comparing dance to exercise [15, 16, 67]. The result was in favor of dance (0.56 points, CI -0.00 to 1.11, p = 0.05) with no heterogeneity (P = 0.73, $I^2$ = 0%).

**FOG-Q.** The effects of dance on freezing of gait (Fig 5.3) were analyzed by combining two studies (n = 52) that compared dance to exercise [15, 16]. There was no significant effect (0.56, CI -0.81 to 1.93, p = 0.42) and there was no evidence of heterogeneity (P = 0.65, $I^2$ = 0%).

**TUG.** The effects of dance on functional mobility measured using the TUG (Fig 5.4) were assessed by combining two studies (n = 52) that compared dance to exercise [15, 16]. The results favored dance (-1.15, CI -2.03 to -0.27, p = 0.01) and there was no evidence of heterogeneity (P = 0.67, $I^2$ = 0%).

**PDQ-39.** The effects of dance on health-related QOL measured using the PDQ-39 (Fig 5.5) were analyzed by combining two studies (n = 77) that compared dance to another form of physical activity. The results favored dance (-4.50 points, CI -7.95 to -1.04, p = 0.01) with very little evidence of heterogeneity (P = 0.31, $I^2$ = 3%) [16, 24].

## Discussion

The primary objective of this review was to evaluate evidence for the efficacy of dance in improving motor impairments, non-motor symptoms, and QOL in people living with PD. Overall, the

### 5.1 MDS-UPDRS III

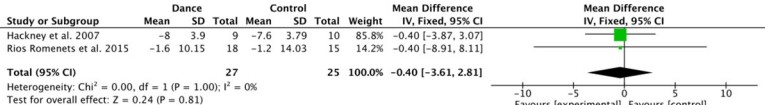

### 5.2 Balance

### 5.3 FOG-Q

### 5.4 TUG (sec)

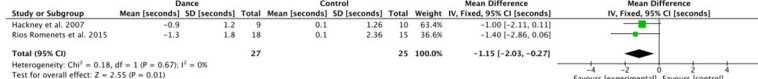

### 5.5 PDQ-39

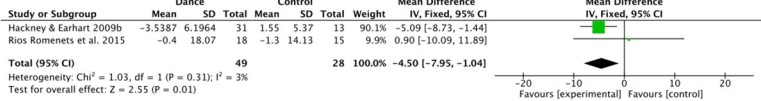

**Fig 5. Meta-analysis results of dance vs. active control.**

evidence suggests that dance can have a positive impact in those living with mild to moderate PD, with the results most strongly supporting its ability to manage motor symptom severity in comparison to usual care and to improve balance and functional mobility more effectively than usual care or another form of physical activity. The study selection criteria and subsequently the included studies resulted in only one meta-analysis on an outcome measuring a non-motor symptom (i.e., symptoms of depression measured using the BDI-II), with the results favoring dance with moderate heterogeneity. The qualitative synthesis similarly highlighted the need for more research in this area to firmly determine the existence and strength of dance-related benefits for non-motor symptoms, which could reveal further mechanisms for improving QOL.

The secondary objectives of this review were to assess the methodological quality of included studies and to inform the direction of future research. In the past four to five years, an increasing number of research groups have begun investigating the impact of dance in people with PD using an RCT design [16, 33, 34, 36, 43, 64, 65], and methodological quality has improved in some areas, such as randomization procedures, while largely remaining the same in others. As healthcare providers increasingly turn to social prescription and arts-based interventions, more high quality, properly powered RCTs will be needed to fully characterize the effects of dance on motor symptoms, to confirm its effects on non-motor symptoms, and to determine which program elements (e.g., dance styles, intensities) and mechanisms of change are most important for improving QOL.

## Motor symptoms

Our meta-analysis supports the idea that dance can improve motor symptom severity more effectively than usual care [14, 17, 43, 65]. Given the progressive nature of PD, it is of great clinical importance that dance has the potential to modify disease progression, which could have implications for outcomes related to disability and QOL. The qualitative synthesis showed that various types of dance can lead to improvements in motor symptoms as measured by the MDS-UPDRS motor subscale as well as or better than other forms of exercise or physiotherapy over the course of three to six months [15, 66]; however, the results of the meta-analysis of two studies comparing dance to an active control did not suggest that dance is more effective than other forms of physical activity [15, 16]. Among the five studies that used the MDS-UPDRS II, which measures motor experiences of daily living, only Duncan and Earhart [62] showed improvement after 12 and 24 months of tango, suggesting longer duration interventions may be necessary before participants begin to perceive changes in motor symptoms.

The meta-analysis also revealed that dance, specifically tango, may be superior at improving balance and functional mobility than other forms of physical activity traditionally available to people with PD [15, 16]. In one of the studies included in this meta-analysis, Rio Romenets et al.'s [16] control group practiced unmonitored, self-directed exercise, and in the other, Hackney et al.'s [15] study, the exercise group spent 50 minutes of the class time seated, while the tango group stood dancing for 60 minutes. Future RCTs should compare the dance programs being assessed to intensity-matched exercise or physiotherapy programs. Volpe et al.'s [66] study compared Irish set dancing to an intensity-matched physiotherapy program and found positive improvements in both groups but better results, although non-significant, in the dance group. Among studies that compared two types of dance, Rocha et al. [36] found tango to improve balance and functional mobility to a greater degree than a mixed-dance program, and Kunkel et al. [33] suggest that teaching several types of ballroom/Latin American dance may have led to null results of motor outcomes by diluting any physical effects. More trials comparing dance programs and principles are warranted, as different techniques may be more effective or may target different motor impairments.

All of the studies included that assessed balance used clinical measures to evaluate change. The specific mechanisms through which dance may improve balance have yet to be elucidated. In 2016, McKay et al. conducted the first study to evaluate automatic postural responses using kinematic and electromyographic outcome measures before and after an adapted tango program, which showed a reduction of forward center of mass displacement and delayed antagonist onset time and duration (measured using EMG) after three weeks of a high volume tango intervention (450 minutes/week) [6]. This uncontrolled study demonstrated that further research investigating the effects of dance on postural control using kinematic and electromyographic outcome measures is warranted and feasible. Surprisingly, only one study reported an outcome measuring fall frequency during the course of an intervention [16]. Given that balance and falls are top research priorities among people living with PD [68], longitudinal studies that measure fall frequency during long-term dance practice are warranted to determine whether improvements in balance in this context translate into improved fall risk.

The findings of the qualitative synthesis and meta-analysis suggest that firm conclusions cannot yet be drawn on the impact of dance on gait variables (e.g., velocity, stride length) and endurance. Sharp and Hewitt [29] posited that dance may not provide intense enough training to improve endurance as measured by the 6MWT. Interestingly, the exclusion of the study with the longest duration intervention (180 minutes of Sardinian folk dance per week) led to a null result in the meta-analysis of the 6MWT, supporting this idea that more intensity may be needed to impact this outcome [65]. A recent study comparing high- versus low-intensity treadmill training found that the lower-intensity training resulted in the greatest improvement in gait speed in participants with PD [69]. Depending on the program design, dance classes can similarly provide light-moderate intensity exercise [6]. Future studies investigating dance should incorporate outcomes that measure intensity (e.g., percentage heartrate reserve, music tempos, etc.) in order to determine if a certain level is needed for positive changes in gait variables and endurance, which are important outcomes to consider given that reduced walking speed may lead to an increased risk of mortality among people with PD [70].

The meta-analysis did not favor dance in comparison to usual care or another form of physical activity in the management of FOG; however, in a trial not eligible for inclusion in the meta-analysis, Irish set dancing was found to improve FOG more than physiotherapy [66]. Additionally, mixed dance was shown to improve FOG more than tango [36] and tango more than waltz/foxtrot [17]. Freezing most commonly occurs when a person with PD is initiating gait, turning, passing through a narrow space, or approaching a target (e.g., a chair) [71]. Dance classes often incorporate learning techniques for turning, require approaching and maneuvering around other dancers, and involve frequent movement initiation and cessation making further exploration in this area justified.

This review also found emerging evidence that dance has the potential to target other motor impairments that impact people with PD, such as turning [34], upper body flexibility [65], and manual dexterity, which may have been reflective of tango's global impact on motor symptoms severity, specifically bradykinesia [14]. Larger, properly powered comparative trials are needed to begin exploring and isolating which elements of dance may specifically target these various motor functions.

## Non-motor symptoms

In comparison to motor symptoms, which were investigated in nearly every RCT, only six studies included outcomes evaluating non-motor symptoms in their analyses [14, 16, 34, 43, 62, 65]. Three studies measured the effects of dance on cognition using the MoCa, and only one saw improvement after twenty-four 90-minute Sardinian folk dance classes [65]. Three studies

measured the impact of dance on dual tasking while walking, with one study finding a positive impact on gait velocity after six and 12 months of tango [14] and one finding improvement in the DT-TUG after three months of tango [16]. Despite the lack of non-motor evaluation in published RCTs, there is evidence supporting tango's potential to impact cognitive function in people living with PD from controlled, non-randomized studies, which demonstrated that it can improve executive function [72] and spatial cognition [20]. There is also evidence that mixed-genre dance classes can lead to large within group effect sizes for cognitive switching and attention [21] and improvement in global cognition and mental rotation ability [22].

Dance is a multidimensional, sensory experience that engages attention, memory, and many other cognitive processes. It has also been identified as the leisure activity most associated with a lower risk of dementia among community-dwelling older adults [73], and a recent review showed that dance can have a positive effect on global cognition in this demographic [74]. Interventions that target cognition are important for people with PD, many of whom develop dementia [75]. Researchers should further develop and test dance programs that can potentially have an impact on cognitive abilities, particularly those that dance would be expected to influence, such as spatial cognition [20].

Mental imagery ability, the cognitive process involved in creating visual, auditory, and kinesthetic images in the mind [76], is another promising tool in PD rehabilitation [77] that may be improved through dance. Dance instructors often use imagery to convey the desired movement quality during teaching and encourage visualization during movement execution, thus it would be interesting to investigate if these skills impart benefits on body schema awareness. Notably, no studies included in this review assessed the impact of dance on proprioception or motor imagery abilities, which are sensory deficits in PD [77]. Future research should also consider the role of action observation and imitation in the context of dance classes, which may be contributing to positive physical and emotional effects [78].

Issues with mental health and fatigue are also common non-motor symptoms of PD that are known to greatly impact QOL [3]. Qualitative studies report that participants experience an improved mood after participating in dance classes [26] and a recent uncontrolled study found dance to lead to a reduction in total mood disturbance in people living with PD as measured by the POMS [23]. The meta-analysis supports this idea that dance can improve symptoms of depression; however, four included studies in this review measured changes in depression and two measured changes in apathy, only one of which reported significant effects [65]. Three studies measured changes in fatigue, with one reporting significant improvement in the tango group when protocol violations were excluded [16]. A recent review investigating the impact of physical activity on non-motor symptoms in PD found that fatigue and apathy were only impacted by aerobic exercise [79], so the majority of the dance interventions included in this review were potentially not intense enough to promote change in these areas [65].

Of the four studies that measured changes in non-motor experiences of daily living using the MDS-UPDRS I, only Duncan & Earhart [62] found significant effects after two years of tango. These results suggest that dance may need to be practiced over a longer period of time before meaningful changes in non-motor symptoms are perceived by participants. Thus, longitudinal studies that both track behavior over longer periods of time and compare dance programs of different intensities are warranted.

## Quality of life

Dance has the potential to improve QOL not only through the management of motor and non-motor symptoms but also through providing social support and a creative outlet. The meta-analysis comparing dance to no intervention did not show an overall effect of dance on

QOL (Fig 4.8), while the meta-analysis comparing dance to an active control favored dance (Fig 5.5). However, only two of the seven studies included in this review that measured changes in QOL showed dance to have positive effects [24, 43]. Lee et al. [43] reported improvement in the PDQL total score, as well as Systemic Symptoms and Social Functioning sub-scores after eight weeks of Qi dance. Hackney & Earhart [24] found 20 tango sessions to have a positive effect on the PDQ-39 SI and Mobility and Social Support sub-scores.

Interestingly, the six studies that did not report significant improvements in QOL measured using the PDQ-39 SI also did not report results for the eight subscales (Mobility, ADLs, Emotional Well-being, Stigma, Social Support, Cognitive Impairment, Communication and Bodily Discomfort) [16, 33, 34, 36, 64, 66]. Moving forward, all studies using the PDQ-39 should report all subscales as some effects may have been left unrevealed in these trials, and it may be discovered that particular styles of dance target specific dimensions of QOL. It has also been suggested that a sample size as large as 394 participants [64] or 52 participants per group [66] is needed to detect a clinically meaningful effect of dance using the PDQ-39. None of the included studies had a sample of this size. Given the ease of administering the PDQ-39, a multi-center trial is feasible and warranted.

In addition to using self-report questionnaires like the PDQ-39, future RCTs should consider including a qualitative element to explore more deeply the impact of dance on QOL from the perspective of the participant. Engaging in an artistic intervention is a complex experience and as such its impact on QOL may not fully be captured through questionnaires. Only one study included in this review included semi-structured interviews to explore the experience of dancing, and participants reported feeling achievement from mastering dance steps and benefiting significantly from interacting with others [33]. Other qualitative studies found dance helped people living with PD redefine their approach to managing symptoms [26], enhance self-efficacy and self-confidence through increased participation [27], and find new pathways of movement that allowed for greater freedom and expression [80]. Houston has advocated for the importance of qualitative research in this context [81] and her 2019 book, *Dancing with Parkinson's*, highlights its relevance. Through interviews and observations, she learned that dancing can help people with PD to learn how to live well with Parkinson's, exert agency in their lives, and experience feelings of beauty, grace, and freedom [82].

Future trials should consider including observations, interviews, or focus groups as outcomes, as qualitative research can enhance trials by optimizing the intervention, contributing an interpretation of quantitative results through triangulation, and revealing the meaning ascribed by participants to dancing [81, 83].

## Quality of evidence

Overall, the methodological quality in the 16 included studies varied across the 10 categories of the risk-of-bias analysis. All included studies had small sample sizes, thus increasing the risk for type II error. There were statistically significant differences between groups at baseline in three of the 16 trials, accounting for 18% of participants analyzed in this review [16, 33, 34]. The baseline differences reported included a higher fall risk, a greater propensity to exercise, older age, higher mean MDS-UPDRS III scores, and a trend toward longer time since diagnosis, all of which could conceivably have an impact on the outcomes of a dance intervention. All of these studies had small sample sizes and were underpowered [16, 33, 34], and the two largest of the three had unclear risk of bias regarding their randomization methods [16, 33], highlighting further the need for larger trials and sound selection methods. In comparing the percentage of risk of bias over time based on the ratings in this review, an improvement in the

randomization methods was found, while allocation concealment methods and reporting remained consistently unclear.

In terms of controlling for the impact of medication on performance during assessment sessions, the majority of the studies tested participants ON their PD medication at a standardized time of day; however, three trials did not describe if or how medication was controlled during the assessment sessions [34, 43, 64]. Additionally, Volpe et al. [66] reported that testing did not always occur at peak dose during medication cycle despite taking place in the same hour. Since the majority of studies used time of day to control for medication-related fluctuations, it is possible that they also were not testing participants consistently during the same period of their medication cycles. There were three trials measuring the effects of tango lasting one year or more that did not report monitoring medication changes or other additional therapies during the course of the intervention; however, participants completed all assessments OFF medication after at least 12 hours of withdrawal so it is unlikely that any improvements were a result of changes in pharmacological treatment [14, 62, 63]. Two of these studies demonstrated the potential for tango to have a disease modifying effect on motor impairments, thus more studies testing participants OFF medication are warranted [14, 62].

With regard to the consistency of co-interventions, most studies reported that participants were instructed to continue with usual care, activities, and exercise outside of the dance intervention; however, in the majority of cases it is not described if or how this was monitored. Only three trials explicitly reported that participants were not engaged in group exercise or therapies outside of the trial [15, 34, 64] and none of the others quantified any additional exercise. Given that Foster et al. [63] showed that dance has the potential to increase activity participation, recording and evaluating changes in participation during the course of interventions in future trials will be worthwhile. This will help determine if increased participation is seen in response to other dance interventions and it will also monitor possible co-interventions, making it easier to confirm whether or not the effects of the dance intervention have truly been isolated. Moreover, if an intervention's primary intention is to improve participant outcomes then increasing participation in physical activity programs could be among its goals. Regular physical activity (i.e., more than 150 minutes per week), a dose offered in only two interventions included in this review [65, 66], is most associated with improved QOL among people living with PD [84].

Future research should also consider the consistency of co-interventions prior to the start of the trial. Lee and colleagues [43] only recruited participants who had no exercise therapy for three months prior to the study, and Michels and colleagues [34] only recruited participants who had not participated in dance interventions for three months prior to the study. Michels et al. [34] additionally controlled for medication changes one month prior to the start of the intervention. These considerations may be important for future trials given that a common outcome of interest, the PDQ-39, asks participants to self-report their experiences based on the past month.

With regard to comparability between trials arms, the majority compared dance to no intervention, with only five trials testing it against an active control and four comparing two different styles of dance. In future studies, more three to four armed trials will be necessary to rigorously compare different styles of dance to other forms of group physical activity and usual care to control for fluctuations in performance related to living with PD. By including both an active and inactive control, future studies can begin to control for performance bias by ensuring that at least two of the groups being compared have received an equal amount of attention and care. Co-interventions (e.g., other exercise activities) must also be quantified and reported and the delivery of assigned dance interventions monitored and recorded in order to mitigate consequences of performance bias [85].

## Conclusions and future directions

Overall, this review supports previous findings that people with mild to moderate PD can benefit from various dance interventions. The evidence at this point most strongly supports dance's ability to manage motor impairments, with more research needed to determine what effects dance may have on non-motor symptoms and how dance may improve QOL.

At present, these results can only be generalized to individuals with mild to moderate idiopathic PD (mean H&Y stage 2.2) who are older adults (mean age 68.4). Future research should investigate the impact of dance on participants living with advanced PD as there have been promising studies demonstrating that dance can have an impact in later stages of the disease. Hackney and Earhart [18] demonstrated that adapted tango can lead to gains in motor impairments and improved QOL in a person with PD who primarily used a wheelchair and was classified at H&Y IV [86]. In another uncontrolled study in 2011, Heiberger et al. found an acute effect of dance on the UPDRS motor subscale in a group of participants with an average H&Y score of 3.8 [87]. These findings suggest that motor impairment can be improved in later stage PD through dance. Additionally, researchers should consider the impact of dancing in those diagnosed with early-onset PD, who, in comparison to typical-onset PD, may experience more severe depression [88], greater perceived stigmatization, a disruption of family life, and worse QOL as measured by the PDQ-39 [89]. Thus, inquiry into whether dance can improve symptoms and QOL in more diverse groups of people living with PD should be explored.

As previous reviews have identified, many of the RCTs investigating dance and PD, namely four out of the five studies included in Sharp and Hewitt's [29] meta-analysis, were conducted by the same research group [14, 15, 17, 18, 24, 62, 63]. Here, the nine studies included in the meta-analysis demonstrated increased diversity in the types of dance techniques studied and trial locations [16, 33, 34, 36, 43]; however, feasibility, acceptability, and results of outcome measures will need to be continually replicated in diverse communities in order to build a more robust body of evidence.

Benefit-cost analyses may also be worthwhile in determining the resources required to produce benefits, and how this compares to similar interventions for people with PD. The specific mechanisms through which dance may improve motor symptoms, in particular postural instability, should also be investigated, and the optimal dosage and intensity for bringing about positive change should be determined. Finally, the experience of dancing among people with PD should be explored using qualitative methods alongside objective measures in interventions to provide a more holistic evaluation of the impact of dance programs. Dance is a complex social experience involving music, learning, and opportunities for self-expression making it difficult to measure its value through quantitative methods alone.

## Supporting information

**S1 File. Inclusion criteria.**
(DOCX)

**S2 File. Full search strategy.**
(DOCX)

**S3 File. Risk of bias judgement tables.**
(DOCX)

**S4 File. PRISMA checklist.**
(DOC)

## Acknowledgments

The authors would like to thank Dr. Matthew Rodger for his feedback on earlier drafts of this review. They would also like to thank the Thouron Award for supporting this research through a grant to Anna Carapellotti.

## Author Contributions

**Conceptualization:** Anna M. Carapellotti, Michail Doumas.

**Data curation:** Anna M. Carapellotti, Rebecca Stevenson, Michail Doumas.

**Formal analysis:** Anna M. Carapellotti, Rebecca Stevenson, Michail Doumas.

**Funding acquisition:** Anna M. Carapellotti.

**Methodology:** Anna M. Carapellotti, Michail Doumas.

**Supervision:** Michail Doumas.

**Writing – original draft:** Anna M. Carapellotti.

**Writing – review & editing:** Anna M. Carapellotti, Rebecca Stevenson, Michail Doumas.

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
