## [Decision Letter · Decision Letter 0]

9 Mar 2020

PONE-D-20-04189

The efficacy of dance for improving motor impairments, non-motor symptoms, and quality of life in Parkinson’s disease: A systematic review and meta-analysis

PLOS ONE

Dear Ms Carapellotti,

Thank you for submitting your manuscript to PLOS ONE. After careful consideration, we feel that it has merit but does not fully meet PLOS ONE’s publication criteria as it currently stands. Therefore, we invite you to submit a revised version of the manuscript that addresses the points raised during the review process, detailed below.

We would appreciate receiving your revised manuscript by Apr 23 2020 11:59PM. To enhance the reproducibility of your results, we recommend that if applicable you deposit your laboratory protocols in protocols.io, where a protocol can be assigned its own identifier (DOI) such that it can be cited independently in the future. For instructions see: http://journals.plos.org/plosone/s/submission-guidelines#loc-laboratory-protocols

We look forward to receiving your revised manuscript.

Kind regards,

Antony Bayer

Academic Editor

PLOS ONE

Journal Requirements:

2) We note that your literature search was completed in December 2018. Please update your search to include studies published in the last 12 months. Please also assess publication bias within your analysis, or indicate why this was not possible.

Reviewers' comments:

Reviewer's Responses to Questions

**Comments to the Author**

1. Is the manuscript technically sound, and do the data support the conclusions?

Reviewer #1: Yes

Reviewer #2: Partly

2. Has the statistical analysis been performed appropriately and rigorously? 

Reviewer #1: Yes

Reviewer #2: Yes

3. Have the authors made all data underlying the findings in their manuscript fully available?

Reviewer #1: No

Reviewer #2: Yes

4. Is the manuscript presented in an intelligible fashion and written in standard English?

Reviewer #1: Yes

Reviewer #2: Yes

5. Review Comments to the Author

Reviewer #1: Review: “The efficacy of dance for improving motor impairments, non-motor symptoms, and quality of life in Parkinson’s disease: A systematic review and meta-analysis.”

Summary of study

This manuscript presents a systematic review and meta-analysis of studies investigating the potential motor and non-motor benefits of dance for people with Parkinson’s disease (PD). The aims were to evaluate the evidence for the efficacy of dance in improving the above outcomes for people with PD, to assess the methodological quality of studies, and to identify future research directions.

While there are several recent reviews and meta-analyses on this topic, this is a fast-growing area of research and the present review considers a number of more recent studies in addition to the previous evidence.

The review focuses on randomised controlled trials, with 15 studies found to be eligible for inclusion, 8 of which were also included in a meta-analysis. The analysis indicated that dance can improve motor impairment in individuals with mild to moderate PD, but further research is needed to elucidate non-motor effects. The authors offer a number of suggestions for future research in this area, including multi-centre trials, investigation of underlying mechanisms, and effects in different groups of PD patients (advanced and young onset).

This is a clearly written, well-structured review of the current literature on dance for PD, which provides an update following previous reviews and meta-analyses on this topic.

Introduction

The authors refer to a number of existing systematic reviews and meta-analyses on dance for PD. They may also wish to note the following references:

Lotzke, D., Ostermann, T., Bussing, A., 2015. Argentine tango in Parkinson disease – a systematic review and meta-analysis. BMC Neurol. 15. https://doi.org/10.1186/s12883-015-0484-0.

Aguiar, L.P.C., Da Rocha, P.A., Morris, M., 2016. Therapeutic dancing for Parkinson’s disease. Int. J. Gerontol. 10, 64–70. https://doi.org/10.1016/j.ijge.2016.02.002.

Methods

Lines 138 and 154: Please correct “Cochran” to “Cochrane”.

Line 161: “In trials where two types of dance were compared to another active control or no intervention, the means and SDs of the two dance groups’ change scores were pooled” – could the authors please explain why this approach was taken, instead of separately examining the effects for each dance style?

Results

The authors do not consistently note the dance styles concerned when discussing findings (e.g., lines 401, 410) – it would be helpful to include this throughout.

Within the qualitative synthesis section, quality of life and activities of daily living are reported under the heading of “Non-motor symptoms” – it would be more appropriate to discuss these under separate headings as they can be impacted by a range of different symptoms.

Line 211: The authors note that only one study involved a programme led by a dance therapist – this is very interesting and it would be good to know what this is in contrast to; i.e., what was the training/background of the instructors leading the other programmes?

Table 1: The column “Analysis Method” is unclear – what does yes/no refer to here?

Line 252: “Six trials …controlled for medication during the course of the intervention” – please clarify in what way medication was controlled.

Line 333: Please correct “Kunkle” to “Kunkel” (also elsewhere in the manuscript).

Discussion

The discussion is comprehensive and highlights limitations in the existing literature, as well as providing suggestions for future research, such as the need to clarify underlying mechanisms of the benefits of dance in PD (e.g., which elements of dance might target which symptoms). However, it is quite a lengthy discussion and in parts repeats information that is also presented in the results – the authors could consider condensing some of this detail.

Line 644: This is a little unclear – do the authors mean “…reduced walking speed has been identified…”?

Line 662: The discussion of “mental imagery” is introduced in the section on motor outcomes – this would be better discussed in the non-motor section, where it is already considered in more detail.

Lines 687-692: The authors discuss the use of imagery within dance and the potential of dance to improve mental imagery skills. Please see the recent review by Bek et al. below, which discusses action representation mechanisms (motor imagery and action observation) in relation to dance for PD.

Bek, J., Arakaki, A., Lawrence, A., Sullivan, M., Ganapathy, G., Poliakoff, E., 2020. Dance and Parkinson’s: A review and exploration of the role of cognitive representations of action. Neuroscience and Biobehavioral Reviews. 109. DOI: 10.1016/j.neubiorev.2019.12.023

Lines 701-703: Please clarify that the study cited here focuses on PD.

Lines 732-734: The UPDRS-II scale (motor experiences of daily living) would be more appropriately discussed within the motor outcomes section rather than the QoL section where it is currently reported.

Conclusions and future directions

Line 817: “non-motor symptoms like cognition and depression” – this needs rephrasing as “cognition” is not a symptom.

Line 843: “…how this compares to other comparable interventions…” – this should be rephrased.

Reviewer #2: Overall the paper was difficult to follow given that there were multiple end-points reviewed in both the meta-analysis and the qualitative summary. From overall review of the paper the overwhelming impression was that whilst there are a number of RCT’s identifying the impact of dance on motor and non-motor symptoms in PD the trials are highly heterogenous in terms of the type of dance intervention (tango/ foxtrot etc.), the intensity of intervention (eg. Total number of hours per week), duration of intervention, methodology (eg. Dance versus usual care or dance versus alternative intervention) and in outcome measures. In addition, most studies were of small sample size and some underpowered.

I felt the authors were able to illustrate the level of bias across studies well with the use of figures 2 and 3. The authors also provided a complementary narrative of bias. There was a good description of dropout rates and reasons for dropouts from trials.

The reporting of level of heterogenicity (I2) with results in the meta-analysis was appropriate however in the discussion and conclusion section of the paper this was not highlighted. As a consequence, statements that commented on the fact that statistically significant results had been observed were not then qualified with a comment that this may be difficult to interpret in the context of the level of heterogenicity between trial designs. For example, [line 569] that ‘overall evidence suggest dance can have a positive impact in mild to moderate PD, with the results most strongly supporting its ability to manage motor symptom severity in comparison to usual care and improve balance more effectively than other forms of physical activity’: this is in the context of an I2 of 90% for MDS-UPDRS and 78% for balance and limits ability to make bold conclusions.

It is commented that heterogenicity values are reported but not used to exclude trials: given the number of trials that are still included despite the degree of heterogenicity this is not reflected in the discussion section when making claims about the benefits of dance in PD.

The authors have addressed their primary objective and provided narrative about insufficient data being available to draw conclusions with respect to the impact of dance on non-motor symptoms. The authors’ section on the quality of evidence is excellent in outlining trial heterogenicity and provides constructive advice to guide the design of future clinical trials in order to reduce bias and investigate relevant clinical parameters. The authors’ conclusion section is concise and insightful.

Suggestions:

As per PRISMA: Was there a review protocol which should be commented on in the methods section?

Line 138- Cochrane (spelling)

Line 148- All trials deemed to have a high risk of performance bias therefore excluded from risk of bias assessment: was it appropriate to do this given as this will affect the overall level of bias? This will undoubtedly affect the validity of results. This should also be mentioned in your conclusions.

Line 154- Cochrane (spelling)

Line 211- Only one trial led by a dance therapist: does this infer that the other trials were not led in a structured approach by a trained professional? If so a brief comment about who delivered the dance training in other trials would be appropriate.

Line 242- You have commented that ‘only’ three out of the fifteen included RCTs had statistically significant differences between groups at baseline. This is still 20% of the trials. It would be helpful to support this comment by looking at the total number of patients in these three trials and establishing how this reflects across the total number of participants across trials. I also think this should be commented on in the ‘Quality of Evidence’ section.

Line 254- Four studies instructed participants to continue with regular exercise outside of intervention. Has the level of exercise outside of intervention been quantified? If so, a comment on additional level of exercise should be made as this may augment any observed benefits from dance therapy.

Line 464- Clinical global impression of change: It should be specified if in this study the examiner was blinded to intervention and if so how this was achieved?

Line 484- You have commented that the heterogenicity values are reported but not used to exclude trials: is this appropriate given the degree of heterogenicity and the data on bias?

6. PLOS authors have the option to publish the peer review history of their article (what does this mean?). If published, this will include your full peer review and any attached files.

Reviewer #1: No

Reviewer #2: No

---

## [Author Response · Author response to Decision Letter 0]

8 May 2020

8 May 2020

Dear Professor Bayer, 

Please find uploaded into the PLOS ONE editorial manager the revision of our manuscript PONE-D-20-04189 “The efficacy of dance for improving motor impairments, non-motor symptoms, and quality of life in Parkinson’s disease: A systematic review and meta-analysis.” 

We would like to firstly thank the reviewers for their insightful questions and suggestions. The feedback led to considerable changes to the manuscript, which we believe has strengthened the article and its findings. 

Below you will find our Response to Reviewers, which addresses each point raised by the reviewers and references the changes made in the track changed version of the manuscript. We hope that you are satisfied with the changes and find the manuscript now acceptable for publication in PLOS ONE. 

Thank you also for the small extension requested in light of the Covid-19 crisis. We appreciate your understanding and flexibility and look forward to your response. 

Kind regards, 

Anna Carapellotti 

PhD Student 

School of Psychology 

Queen’s University Belfast

18-30 Malone Road 

Belfast

BT9 6HJ

Tel (mobile): +44 (0) 7454005147 

Email: acarapellotti01@qub.ac.uk

 

Thank you for providing the templates. We realized there was an issue with the reference formatting and heading levels. We have adjusted these issues. Please let us know if we have missed anything else.

2) We note that your literature search was completed in December 2018. Please update your search to include studies published in the last 12 months. 

We thank the editor for this recommendation. We have updated the search through March 2020 week 4. This yielded the inclusion of one further trial that investigated a novel dance intervention in people living with Parkinson’s, Sardinian Folk dance. This trial was included in both the qualitative synthesis and meta-analysis: 

Solla P, Cugusi L, Bertoli M, Cereatti A, Della Croce U, Pani D, et al. Sardinian folk dance for individuals with Parkinson’s disease: A randomized controlled pilot trial. J Altern Complement Med. 2019;25(3):305-316. doi: 10.1089/acm.2018.0413.

Please also assess publication bias within your analysis, or indicate why this was not possible.

We have outlined measures taken to reduce the risk of publication bias (see lines 329-336). Statistical techniques used to assess publication/reporting bias could not be used in this review because none of our meta-analyses included more than 10 studies combined. We have made this clear in the results section (see lines 594-596). 

Review Comments to the Author

Reviewer #1: Review: “The efficacy of dance for improving motor impairments, non-motor symptoms, and quality of life in Parkinson’s disease: A systematic review and meta-analysis.”

Summary of study

This manuscript presents a systematic review and meta-analysis of studies investigating the potential motor and non-motor benefits of dance for people with Parkinson’s disease (PD). The aims were to evaluate the evidence for the efficacy of dance in improving the above outcomes for people with PD, to assess the methodological quality of studies, and to identify future research directions.

While there are several recent reviews and meta-analyses on this topic, this is a fast-growing area of research and the present review considers a number of more recent studies in addition to the previous evidence.

The review focuses on randomised controlled trials, with 15 studies found to be eligible for inclusion, 8 of which were also included in a meta-analysis. The analysis indicated that dance can improve motor impairment in individuals with mild to moderate PD, but further research is needed to elucidate non-motor effects. The authors offer a number of suggestions for future research in this area, including multi-centre trials, investigation of underlying mechanisms, and effects in different groups of PD patients (advanced and young onset).

This is a clearly written, well-structured review of the current literature on dance for PD, which provides an update following previous reviews and meta-analyses on this topic.

We would like to thank the reviewer for their positive comments. We have addressed the issues raised below:

Introduction

1. The authors refer to a number of existing systematic reviews and meta-analyses on dance for PD. They may also wish to note the following references:

Lotzke, D., Ostermann, T., Bussing, A., 2015. Argentine tango in Parkinson disease – a systematic review and meta-analysis. BMC Neurol. 15. https://doi.org/10.1186/s12883-015-0484-0.

Aguiar, L.P.C., Da Rocha, P.A., Morris, M., 2016. Therapeutic dancing for Parkinson’s disease. Int. J. Gerontol. 10, 64–70. https://doi.org/10.1016/j.ijge.2016.02.002.

We would like to thank the reviewer for this suggestion. We have added these references. Please see lines 81-83.

Methods

2. Lines 138 and 154: Please correct “Cochran” to “Cochrane”.

This error has been corrected throughout the manuscript. Please see lines 145 and 163.

3. Line 161: “In trials where two types of dance were compared to another active control or no intervention, the means and SDs of the two dance groups’ change scores were pooled” – could the authors please explain why this approach was taken, instead of separately examining the effects for each dance style?

Thank you for requesting this clarification. We took this approach in line with previous meta-analyses because our aim was to compare dance to either no intervention or an active control, rather than to compare two types of dance (Sharp & Hewitt, 2014). Moreover, it was only in two studies where this was necessary (Hackney & Earhart, 2009a, 2009b), and the two styles of dance investigated in these trials were tango and waltz/foxtrot, which are both partnered dance styles. Moreover, the classes were taught by the same instructor. We have included information about this in lines 174-76. 

Results

4. The authors do not consistently note the dance styles concerned when discussing findings (e.g., lines 401, 410) – it would be helpful to include this throughout.

We agree with the reviewer and we have made this change throughout the results section.

5. Within the qualitative synthesis section, quality of life and activities of daily living are reported under the heading of “Non-motor symptoms” – it would be more appropriate to discuss these under separate headings as they can be impacted by a range of different symptoms.

We would like to thank the reviewer for pointing this out. Because PLOS ONE only permits three levels of subject headings, we realized that it is not possible to include the headings Motor Symptoms, Non-Motor Symptoms, and Quality of Life. We have thus removed the headings “Motor symptoms” (line 343) and “Non-motor symptoms” (line 504). We hope that this resolves the issue. 

6. Line 211: The authors note that only one study involved a programme led by a dance therapist – this is very interesting and it would be good to know what this is in contrast to; i.e., what was the training/background of the instructors leading the other programmes?

We have added a section that details the qualifications and supervision of the dance instructors in each trial. Please see lines 231 through 236 for details. 

7. Table 1: The column “Analysis Method” is unclear – what does yes/no refer to here?

The yes/no refers to whether or not “Intention-to-treat” was used as the method of analysis. We have updated the table to reflect this (please see line 237). 

8. Line 252: “Six trials …controlled for medication during the course of the intervention” – please clarify in what way medication was controlled.

We have now clarified this issue, please see lines 280-291. 

9. Line 333: Please correct “Kunkle” to “Kunkel” (also elsewhere in the manuscript).

This error has been corrected throughout the manuscript. 

Discussion

10. The discussion is comprehensive and highlights limitations in the existing literature, as well as providing suggestions for future research, such as the need to clarify underlying mechanisms of the benefits of dance in PD (e.g., which elements of dance might target which symptoms). However, it is quite a lengthy discussion and in parts repeats information that is also presented in the results – the authors could consider condensing some of this detail.

We would like to thank the reviewer for this observation and suggestion. We agreed that it was repetitive and have trimmed the discussion to make it more concise. Please see line 694 through end of paper. 

11. Line 644: This is a little unclear – do the authors mean “…reduced walking speed has been identified…”?

Apologies for the confusion. Yes, we meant “reduced walking speed has been identified…” This sentence has been edited for clarity. Please see line 793.

12a. Line 662: The discussion of “mental imagery” is introduced in the section on motor outcomes – this would be better discussed in the non-motor section, where it is already considered in more detail.

12b. Lines 687-692: The authors discuss the use of imagery within dance and the potential of dance to improve mental imagery skills. Please see the recent review by Bek et al. below, which discusses action representation mechanisms (motor imagery and action observation) in relation to dance for PD.

Bek, J., Arakaki, A., Lawrence, A., Sullivan, M., Ganapathy, G., Poliakoff, E., 2020. Dance and Parkinson’s: A review and exploration of the role of cognitive representations of action. Neuroscience and Biobehavioral Reviews. 109. DOI: 10.1016/j.neubiorev.2019.12.023

We would like to thank the reviewer for these suggestions. The entire discussion of mental imagery has been moved to the non-motor section and we have referenced Bek and colleagues’ newly published review. Please see lines 843-852.

13. Lines 701-703: Please clarify that the study cited here focuses on PD.

This issue has now been clarified (see lines 857). 

14. Lines 732-734: The UPDRS-II scale (motor experiences of daily living) would be more appropriately discussed within the motor outcomes section rather than the QoL section where it is currently reported.

We have now moved the discussion of subscale II (motor experiences of daily living) to the Motor Outcomes section (see lines 741-743). In line with this change we moved the discussion of subscale I (non-motor experiences of daily living) to the Non-motor outcomes section (see lines 867-875). 

Conclusions and future directions

15. Line 817: “non-motor symptoms like cognition and depression” – this needs rephrasing as “cognition” is not a symptom.

Thank you for pointing out this error. We deleted the examples of non-motor symptoms here because it felt repetitive (line 1026). We realized that this issue of calling cognition a symptom also needed to be addressed elsewhere and have thus changed “cognition” to cognitive impairment/function throughout the paper. 

16. Line 843: “…how this compares to other comparable interventions…” – this should be rephrased.

We have rephrased this sentence (see line 1053).

Reviewer #2: 

1. Overall the paper was difficult to follow given that there were multiple end-points reviewed in both the meta-analysis and the qualitative summary. From overall review of the paper the overwhelming impression was that whilst there are a number of RCT’s identifying the impact of dance on motor and non-motor symptoms in PD the trials are highly heterogenous in terms of the type of dance intervention (tango/ foxtrot etc.), the intensity of intervention (eg. Total number of hours per week), duration of intervention, methodology (eg. Dance versus usual care or dance versus alternative intervention) and in outcome measures. In addition, most studies were of small sample size and some underpowered. 

I felt the authors were able to illustrate the level of bias across studies well with the use of figures 2 and 3. The authors also provided a complementary narrative of bias. There was a good description of dropout rates and reasons for dropouts from trials.

The reporting of level of heterogenicity (I2) with results in the meta-analysis was appropriate however in the discussion and conclusion section of the paper this was not highlighted. As a consequence, statements that commented on the fact that statistically significant results had been observed were not then qualified with a comment that this may be difficult to interpret in the context of the level of heterogenicity between trial designs. For example, [line 569] that ‘overall evidence suggest dance can have a positive impact in mild to moderate PD, with the results most strongly supporting its ability to manage motor symptom severity in comparison to usual care and improve balance more effectively than other forms of physical activity’: this is in the context of an I2 of 90% for MDS-UPDRS and 78% for balance and limits ability to make bold conclusions.

It is commented that heterogenicity values are reported but not used to exclude trials: given the number of trials that are still included despite the degree of heterogenicity this is not reflected in the discussion section when making claims about the benefits of dance in PD.

The authors have addressed their primary objective and provided narrative about insufficient data being available to draw conclusions with respect to the impact of dance on non-motor symptoms. The authors’ section on the quality of evidence is excellent in outlining trial heterogenicity and provides constructive advice to guide the design of future clinical trials in order to reduce bias and investigate relevant clinical parameters. The authors’ conclusion section is concise and insightful.

We would like to thank the reviewer for their very positive evaluation of our paper. Regarding the first sentence of the review stating that ‘Overall the paper was difficult to follow given that there were multiple end-points reviewed in both the meta-analysis and the qualitative summary’ we agree that there are multiple end-points in the paper, but this reflects the nature of the reviewed studies. In this revision, we have tried to consolidate these end-points as best as we can in line with both reviewers’ suggestions.

Suggestions:

2. As per PRISMA: Was there a review protocol which should be commented on in the methods section?

A statement regarding the review protocol has been added (please see lines 125-6).

3. Line 138 and 154- Cochrane (spelling)

We would like to thank the reviewer for pointing out this error. This error has been corrected throughout the manuscript. Please see lines 145 and 163.

4. Line 148- All trials deemed to have a high risk of performance bias therefore excluded from risk of bias assessment: was it appropriate to do this given as this will affect the overall level of bias? This will undoubtedly affect the validity of results. This should also be mentioned in your conclusions.

We agree that even though we have pointed out the high risk of performance bias, it would be better to include this result in the figures to provide a better illustration of the risk of bias across trials. We have done this and we have also discussed this issue in the results and discussion section. Please see lines 157, 242, 1013, Fig. 2, and Fig. 3.

5. Line 211- Only one trial led by a dance therapist: does this infer that the other trials were not led in a structured approach by a trained professional? If so a brief comment about who delivered the dance training in other trials would be appropriate.

We have added a section that details the qualifications and supervision of the dance instructors in each trial. Please see lines 231 through 236 for details.

6. Line 242- You have commented that ‘only’ three out of the fifteen included RCTs had statistically significant differences between groups at baseline. This is still 20% of the trials. It would be helpful to support this comment by looking at the total number of patients in these three trials and establishing how this reflects across the total number of participants across trials. I also think this should be commented on in the ‘Quality of Evidence’ section.

We agree that the use of the word “only” here was incorrect. We have omitted it, and we have indicated the number of participants analyzed in these three trials and how this reflects across the total number of participants (see line 270). This has also been commented on in the Quality of Evidence section (see lines 949-957). 

7. Line 254- Four studies instructed participants to continue with regular exercise outside of intervention. Has the level of exercise outside of intervention been quantified? If so, a comment on additional level of exercise should be made as this may augment any observed benefits from dance therapy.

No, the level of exercise outside of the intervention was not quantified in any of the studies. We have now commented on the lack of monitoring in the results section (lines 292-305) and the potential consequences of this in the discussion (lines 983-1018). 

8. Line 464- Clinical global impression of change: It should be specified if in this study the examiner was blinded to intervention and if so how this was achieved?

Thank you for pointing this out, as lack of blinding for this outcome in particular is noted as a limitation in this study. We have included this information in our review. Please see line 570.

9. Line 484- You have commented that the heterogenicity values are reported but not used to exclude trials: is this appropriate given the degree of heterogenicity and the data on bias?

This is an important issue, which led us to look at the data again and to discover that there was an error in the way we calculated standard deviation change scores. Due to this error, high values of heterogeneity were observed. The error has been corrected in both the meta-analysis figures (Figs 4 and 5) and the results section (beginning at line 583). The values have also been updated due to the addition of one more study in the meta-analysis (Solla et al., 2019). 

Moreover, we have detailed how the change standard deviations were calculated in our methods section for clarity (please see lines 1170-172). High values of heterogeneity remained in one meta-analysis (6MWT comparing dance to no intervention, Fig 4.7); however, given the small number of studies included in the meta-analysis, and the fact that clinical diversity was controlled for, it was not seen as appropriate to exclude the study (see lines 626-631). We did conduct a sensitivity analysis on this outcome given the high heterogeneity value and reported the results (638-643).

All files were uploaded into PACE to meet the PLOS requirements.

---

## [Decision Letter · Decision Letter 1]

2 Jun 2020

PONE-D-20-04189R1

The efficacy of dance for improving motor impairments, non-motor symptoms, and quality of life in Parkinson’s disease: A systematic review and meta-analysis

PLOS ONE

Dear Dr. Carapellotti,

Thank you for submitting your revised manuscript to PLOS ONE and for your careful attention to the previous reviewer comments. After careful consideration, we feel that it has considerable merit but would benefit from some minor changes to respond to the suggestions below. I also spotted another typo in the abstract (line 22) and perhaps the whole manuscript would benefit from a final check. Therefore it does not fully meet PLOS ONE’s publication criteria as it currently stands and we invite you to submit a revised version of the manuscript that addresses these points.

We look forward to receiving your revised manuscript.

Kind regards,

Antony Bayer

Academic Editor

PLOS ONE

Reviewers' comments:

Reviewer's Responses to Questions

**Comments to the Author**

1. If the authors have adequately addressed your comments raised in a previous round of review and you feel that this manuscript is now acceptable for publication, you may indicate that here to bypass the “Comments to the Author” section, enter your conflict of interest statement in the “Confidential to Editor” section, and submit your "Accept" recommendation.

Reviewer #1: All comments have been addressed

Reviewer #2: All comments have been addressed

2. Is the manuscript technically sound, and do the data support the conclusions?

Reviewer #1: Yes

Reviewer #2: Yes

3. Has the statistical analysis been performed appropriately and rigorously? 

Reviewer #1: Yes

Reviewer #2: Yes

4. Have the authors made all data underlying the findings in their manuscript fully available?

Reviewer #1: Yes

Reviewer #2: Yes

5. Is the manuscript presented in an intelligible fashion and written in standard English?

Reviewer #1: Yes

Reviewer #2: Yes

6. Review Comments to the Author

Reviewer #1: Thank you for addressing all the points raised in the reviews - you have provided a clear and comprehensive revision and improved the overall quality of the manuscript.

Reviewer #2: The revisions to the article have overall contributed to a more measured and transparent systematic review and meta-analysis. The formatting and editing have resulted in a paper that is far less repetitive and clearer to comprehend. In addition, clarifying outcomes in relation to intervention/ dance style throughout the article has provided added clarity. The quality of evidence section provides an excellent overview of the limitations of data interpretation in view of small sample sizes, inclusion of underpowered studies, statistically significant differences in some studies baseline characteristics, the randomisation process, impact of medication and consistency of co-intervention. In addition the conclusions section provides an excellent summary and very valid recommendations for the methods future research could employ.

The authors have met both their primary and secondary objectives. The authors have also been candid in their descriptions of limitations of generalisability with reference to multiple areas of bias, inclusion of relevant heterogenicity scores, comments on the significant drop-out rate for studies and explanations provided for reasons for participant withdrawal and comments on the incomplete retrieval of identified research.

My only further suggestions are:

Line 22: spelling correction 'mon' to non-motor

Line 714: spelling correction 'met' to meta-analysis

Line 715: I think it is inaccurate to claim 'dance can have an impact on gait velocity, stride length and endurance' when the qualitative synthesis showed variable outcomes and when the meta-analysis observed no significant effect on forward or backward velocity in the dance versus no intervention arms, no effect on stride length in the dance versus no intervention arms and whilst the 6MWT favoured dance, the caveat is that the heterogenicity value for those studies was 84%. It would be more accurate to suggest that the impact of dance on gait velocity, stride length and endurance is not robust enough to draw firm conclusions and should be a focus of future research.

7. PLOS authors have the option to publish the peer review history of their article (what does this mean?). If published, this will include your full peer review and any attached files.

Reviewer #1: No

Reviewer #2: No

---

## [Author Response · Author response to Decision Letter 1]

13 Jul 2020

13 July 2020

Dear Professor Bayer, 

Please find uploaded into the PLOS ONE editorial manager the second revision of our manuscript PONE-D-20-04189 “The efficacy of dance for improving motor impairments, non-motor symptoms, and quality of life in Parkinson’s disease: A systematic review and meta-analysis.” 

We would like to firstly thank the reviewers again for their insightful suggestions. Below you will find our Response to Reviewers, which addresses each of the final points raised and references the changes made in the track changed version of the manuscript. As suggested, we gave the manuscript a final read and have corrected all typos, grammatical, and formatting errors and inconsistencies. We realized that the “Criteria for considering studies” was not presented in the traditional PICO format, so this was also corrected (lines 110-140). 

We hope that you are satisfied with the changes and find the manuscript now acceptable for publication in PLOS ONE. 

Kind regards, 

Anna Carapellotti 

PhD Student 

School of Psychology 

Queen’s University Belfast

18-30 Malone Road 

Belfast

BT9 6HJ

Tel (mobile): +44 (0) 7454005147 

Email: acarapellotti01@qub.ac.uk

 

6. Review Comments to the Author

Reviewer #1: Thank you for addressing all the points raised in the reviews - you have provided a clear and comprehensive revision and improved the overall quality of the manuscript.

Thank you once again for your suggestions which undoubtedly improved the overall quality of the manuscript. 

Reviewer #2: The revisions to the article have overall contributed to a more measured and transparent systematic review and meta-analysis. The formatting and editing have resulted in a paper that is far less repetitive and clearer to comprehend. In addition, clarifying outcomes in relation to intervention/ dance style throughout the article has provided added clarity. The quality of evidence section provides an excellent overview of the limitations of data interpretation in view of small sample sizes, inclusion of underpowered studies, statistically significant differences in some studies baseline characteristics, the randomisation process, impact of medication and consistency of co-intervention. In addition the conclusions section provides an excellent summary and very valid recommendations for the methods future research could employ.

The authors have met both their primary and secondary objectives. The authors have also been candid in their descriptions of limitations of generalisability with reference to multiple areas of bias, inclusion of relevant heterogenicity scores, comments on the significant drop-out rate for studies and explanations provided for reasons for participant withdrawal and comments on the incomplete retrieval of identified research.

My only further suggestions are:

Line 22: spelling correction 'mon' to non-motor 

Line 714: spelling correction 'met' to meta-analysis

Thank you for pointing out these typos. They have been corrected and the manuscript was proofread a final time for other typos, grammar, and formatting errors. 

Line 715: I think it is inaccurate to claim 'dance can have an impact on gait velocity, stride length and endurance' when the qualitative synthesis showed variable outcomes and when the meta-analysis observed no significant effect on forward or backward velocity in the dance versus no intervention arms, no effect on stride length in the dance versus no intervention arms and whilst the 6MWT favoured dance, the caveat is that the heterogenicity value for those studies was 84%. It would be more accurate to suggest that the impact of dance on gait velocity, stride length and endurance is not robust enough to draw firm conclusions and should be a focus of future research.

We agree with the reviewer and have updated this point accordingly (lines 740 through 755).

---

## [Editor Report · Decision Letter 2]

15 Jul 2020

The efficacy of dance for improving motor impairments, non-motor symptoms, and quality of life in Parkinson’s disease: A systematic review and meta-analysis

PONE-D-20-04189R2

Dear Dr. Carapellotti,

Thank you for your further manuscript and careful revision. We’re pleased to inform you that your manuscript has been judged scientifically suitable for publication and will be formally accepted for publication once it meets all outstanding technical requirements.

Kind regards,

Antony Bayer

Academic Editor

PLOS ONE
---

## [Editor Report · Acceptance letter]

23 Jul 2020

PONE-D-20-04189R2 

The efficacy of dance for improving motor impairments, non-motor symptoms, and quality of life in Parkinson’s disease: A systematic review and meta-analysis 

Dear Dr. Carapellotti:

I'm pleased to inform you that your manuscript has been deemed suitable for publication in PLOS ONE. Congratulations! Your manuscript is now with our production department. 

Kind regards, 

on behalf of

Professor Antony Bayer 

Academic Editor

PLOS ONE